# Effects of Ultrasonication Time on the Properties of Polyvinyl Alcohol/Sodium Carboxymethyl Cellulose/Nano-ZnO/Multilayer Graphene Nanoplatelet Composite Films

**DOI:** 10.3390/nano10091797

**Published:** 2020-09-10

**Authors:** Tengteng Ji, Rong Zhang, Xiaorong Dong, Dur E Sameen, Saeed Ahmed, Suqing Li, Yaowen Liu

**Affiliations:** 1College of Food Science, Sichuan Agricultural University, Ya’an 625014, China; jtt5881@163.com (T.J.); zhangronglzy@163.com (R.Z.); dxr1175752318@163.com (X.D.); sameen0388@gmail.com (D.E.S.); saeedahmedM1993@gmail.com (S.A.); 2School of Materials Science and Engineering, Southwest Jiaotong University, Chengdu 610031, China

**Keywords:** carboxymethyl cellulose, zinc oxide, multilayer graphene nanoplatelets

## Abstract

Ultrasonication-assisted solution casting was used to prepare polyvinyl alcohol (PVA)/sodium carboxymethyl cellulose (CMC)/nano-ZnO/multilayer graphene nanoplatelet (*x*GnP) composite films; the performances (mechanical properties, water vapor permeability (WVP), biodegradability and antibacterial activity) of these films were investigated as a function of the ZnO NPs:*x*GnP mass ratio and ultrasonication time. Intermolecular interactions among ZnO NPs, *x*GnP and the PVA/CMC matrix were shown to improve WVP, while X-ray diffraction and scanning electron microscopy analyses revealed that the internal reticular structure of ultrasound-treated PVA/CMC/ZnO NPs/*x*GnP composite films was in a fluffier state than that of the untreated composite films and the PVA/CMC film. The incorporation of ZnO NPs and *x*GnP into the composite film reduced its tensile strength and elongation at break, and increased antibacterial activity and biodegradability. In addition, we carried out the experiment of strawberry preservation and measured weight loss ratio, firmness, content of total soluble solids and titration acid. Finally, the composite film of 7:3 had the best preservation effect on strawberries. Thus, the obtained results paved the way to develop novel biodegradable composite films with antimicrobial activity for a wide range of applications.

## 1. Introduction

Nowadays, most food packaging materials mainly consist of petrochemical-based polymers because they are convenient, low-cost and have excellent barrier performances [1]. However, it is commonly accepted that those nonbiodegradable materials will cause irreversible environmental contaminations both in the long- and short-term, and are harmful to public health [2]. Therefore, it is crucial to develop degradable, antibacterial and cheap polymer films for food packaging.

Some of the materials that have been used for the preparation of packaging materials which have biodegradable properties are starch, PVA and polylactic acid (PLA) [3]. Among these, PVA is a non-toxic, highly crystalline and water-soluble polymer that has good film-forming and high hydrophilic properties [2]. PVA can also supply hydroxyl groups on the surface of nanofibers. The hydroxyl groups can also serve to adsorb heavy metal ions [4]. Because of the hydrophilic hydroxyl group (–OH) in the molecule, the water resistance of PVA film can be improved if the hydroxyl group can be closed properly and connected to the water-resistant group. PVA is often combined with cellulose and its derivatives, such as carboxymethyl cellulose (CMC), to prepare composites. CMC has abundant carboxyl groups and is therefore sensitive to water, aiding its absorption by PVA, while strong hydrogen bonding between the hydroxyl groups of PVA and CMC due to blending results in remarkable mechanical properties. PVA/CMC blends are considered to be a useful matrix for improving the solid polymer, and have many potential applications [5,6]. However, pure PVA/CMC composite films feature low water resistance and have no resistance to pathogens. Coincidentally, many studies have shown that when inorganic NPs are added to a polymer, their antibacterial properties are greatly improved. Zhong et al. [7] attempted to fabricate a multi-functional ZnO NPs/CS composite with enhanced biosafety in an efficient and low-cost way, which has potential applications as an antibacterial, UV-shielding or dye water-treatment agent. Currently, ZnO NPs is one of the five zinc compounds that are listed as generally recognized as safe (GRAS) materials by the U.S. Food and Drug Administration [8]. ZnO NPs possess many advantages, such as being non-toxic, allowing cheap and easy synthesis of semiconductor materials, having high photocatalytic activity, good chemical stability and resistance to light corrosion, and are easily doped, and so on [9], whereas their mechanical properties experience deterioration due to the NP surface effect and interfacial action [9].

In addition, numerous metal oxides and NPs have been compounded with graphene sheet layers to obtain superior composites, as the addition of appropriate amounts of graphene to polymers can significantly improve their mechanical properties through the effects of van der Waals forces. Parameswaranpillai et al. [10] have shown that the incorporation of exfoliated graphene nanoplatelets (*x*GnP) into ternary PP/PS/SEBS blends significantly improves the tensile modulus and elongation at break of ternary. The ease of production and high chemical inertness of graphene make it a promising filler, although its widespread use is hindered by its poor dispersibility in both organic and inorganic solvents. In addition, the non-uniform distribution of nanofillers in the polymer matrix may result in mechanical property deterioration and pore formation [11].

Ultrasonication is widely employed in the field of material design and engineering, as its heating, mechanical, super-mixing and cavitation effects facilitate the breakage of the original hydrogen bonds and hydrophobic bonds in the medium, and thus promote the formation and exposure of more reaction centers [12]. In turn, this accelerates chemical reactions and allows macromolecules to recombine and form a variety of nonpolar bonds. Moreover, ultrasonication further improves material solubility, and makes films adopt more compact network structures [13]. The intensity of the effect of using a direct ultrasonication device (probe) in solution is far greater than that of mechanical agitation, as the former allows the particles in the solution to be uniformly distributed on a macroscopic scale, while cavitation aids further particle homogenization on a microscopic scale [14]. Hence, the use of a direct ultrasonication device helps one to avoid nanofiber agglomeration. However, the effect of ultrasonication time on the physicochemical properties of composite films remains underexplored. Herein, to address this knowledge gap, we fabricated PVA/CMC/nano ZnO NPs/multilayer graphene nanoplatelet (*x*GnP) films using a combination of ultrasonication and solution casting methods, and explored the influence of the ZnO NPs:*x*GnP mass ratio and ultrasonication time on film properties.

## 2. Materials and Methods

### 2.1. Materials

PVA (Mw = 7.6 kDa, Mw/Mn = 1.32, 1700 degree of polymerization and 99% hydrolyzation) was procured from Shenzhen Esun Industrial Co., Ltd. (Shenzhen, China). Analytical grade CMC (300–800 mPa·s) was purchased from Shanghai Chemical Agents Co., Ltd. (Shanghai, China) and used as received. ZnO NPs (average size = 30 nm) were purchased from Sigma-Aldrich. *x*GnP (multilayer graphene nanoplatelet, with strong toughness and good antibacterial properties) was sourced from Suzhou Yougao Nanomaterials Co., Ltd.

### 2.2. Preparation of PVA/CMC Films

PVA/CMC films were obtained by solution casting. PVA (10 g) and CMC (1 g) were dissolved in distilled water (200 mL) upon magnetic stirring at 80 °C to afford a PVA/CMC mixed solution. This solution was split into five portions and subjected to an ultrasonic processor (Hielscher UP400S, 400 W, 24 kHz, Labsun China, Shanghai, China), ultrasonicated for 0, 15, 30, 45 and 60 min, and the final PVA/CMC solutions were poured onto plastic plates, and extended to obtain uniform initial films that were oven-dried at 30 °C for 4 h. After the solvent was completely eliminated, the plastic plates with films were further dried overnight at room temperature.

### 2.3. Preparation of PVA/CMC/ZnO NPs/xGnP Composite Films

ZnO NPs and *x*GnP (The total mass of ZnO NPs and *x*GnP is 10% of that of PVA and CMC; mZnO NPs:mG = 10:0, 9:1, 7:3, 5:5, 3:7, 1:9 and 0:10) were suspended in distilled water (200 mL), and the suspension was placed in an ultrasonic bath for 60 min. Next, PVA (10 g) and CMC (1 g) were dissolved in the suspension using magnetic stirring at 80 °C. The obtained solution was processed using the process described for pure PVA/CMC films to obtain PVA/CMC/ZnO NPs/*x*GnP composite films.

### 2.4. Swelling Capacity and Solubility

This process was slightly modified according to the method of Yan et al. [15]. The films were cut into 3 cm × 3 cm pieces for the determination of *solubility* and swelling degree. The pieces were dried at 105 °C to a constant weight to obtain the initial dry mass (*M*_1_). Then, they were placed in 100 mL beakers with 50 mL distilled water covered with plastic wraps and stored at 25 °C for 24 h. Next, the films were dried superficially with filter papers and dried at 105 °C to a constant weight to obtain the final dry mass (*M*_2_). Then, the *solubility* was calculated using the following equation:(1)$$solubility=\;\frac{M_{1}-M_{2}}{M_{1}},$$

The films were put into 50 mL beakers with 30 mL distilled water for 24 h at 25 °C after weighing the films (*M*_1_). The wet films were then dried superficially with filter papers, followed by weighing the wet films (*M*_2_). The *swelling* degree was calculated using the following equation:(2)$$swelling=\;\frac{M_{2}-M_{1}}{M_{1}},\;$$

For each film, three parallel samples of strips were measured, and the average was used as the final result.

### 2.5. Scanning Electron Microscopy (SEM)

Film surface morphology was analyzed by SEM (TOPCON ABT-150 S, COXEM company, Beijing, China). Prior to imaging, the samples were hung on metal grids and coated with gold vapor under vacuum [16].

### 2.6. Attenuated Total Reflectance-Fourier Transform Infrared (ATR-FTIR) Spectroscopy

ATR-FTIR (Prestige-21, Shimazu International Trade (Shanghai) Co., Ltd., Shanghai, China) spectra were recorded over a wavenumber range of 650–4000 cm^−1^ at room temperature, a resolution of 4 cm^−1^, and a scan frequency of 32 s^−1^ to identify major functional groups [17].

### 2.7. X-ray Diffractometry (XRD)

Sample phase composition was determined by powder XRD using a D8 propulsive diffractometer (Bruker, Karlsruhe, Germany) with Cu Kα radiation at 40 kV and 40 mA. XRD spectra were recorded in a 2*θ* range of 5–80°, at a step size of 0.02° and a counting time of 1 s [18].

### 2.8. Differential Scanning Calorimetry (DSC)

DSC (Q200 V24.2 Build 107, New Castle, DE, USA) can be used to determine thermal properties, such as glass transition temperature (*T_g_*), melting temperature (*T_m_*) and crystallization temperature. Herein, the thermal stability of the films was examined using a DSC-50 instrument (Shimadzu) and an empty pan as a reference. The HPe system was heated at an elevated temperature to eliminate previous thermal history, and then cooled at a linear rate before it was heated again. The TA Universal analysis software was used to determine the *T_g_* of the HPe system as it was heated from 50 to 350 °C at a rate of 10 °C·min^−1^ under a nitrogen flow (50 mL·min^−1^) [19].

### 2.9. Density Determination

The composite film was cut into pieces of the same size (10 mm × 100 mm), and a thickness gauge with an accuracy of 0.001 mm (Mitutoyo Absolute, Tester Sangyo Co., Ltd., Tokyo, Japan) was used to measure their thicknesses (*d*) at one point in the center and four points around the center. The result was expressed as the mean of these five values and presented with an accuracy of 0.001 mm. Mass (*m*) was determined using an electronic scale with an accuracy of 0.0001 mg, and combined with the specimen area (*s*), was used to calculate film density (*ρ*) as:(3)$$\rho =\;\frac{m}{s\cdot d},$$

Density measurements were performed three times for each specimen, and the result was expressed as the mean of these measurements [20].

### 2.10. Determination of Water Vapor Permeability (WVP)

A PERME W3/031 water vapor transmittance tester was used for WVP determination (following ASTM E96 (2016) standard) [21]. The composite film was cut into a disk of 33 mm radius, and its thickness (*d*) was measured using a thickness gauge (Mitutoyo Absolute, Tester Sangyo Co., Ltd., Tokyo, Japan) at one point in the center and four points around the center. The result was expressed as the mean of these five values. The samples were put into the water vapor transmittance tester and tested at a temperature of 25 °C and a relative humidity (RH) of 75%. WVP was calculated as:(4)$$WVP=\;\frac{WVTR}{P\left(R_{1}-R_{2}\right)}\;\times x,$$
where *P* is the saturation vapor pressure of water (Pa) at the test temperature (25 °C), R_1_ is the RH in the desiccator, R_2_ is the RH in the permeation cell, and *x* is film thickness (*m*). Under these conditions, the driving force for water vapor permeation equals [*P*(*R*_1_ − *R*_2_)] = 1753.55 Pa.

### 2.11. Determination of Mechanical Properties

The mechanical properties were determined for bubble- and notch-free samples using a tensile tester (HD B609B-S, Haida International Equipment Co., Ltd., Taichung, China) and appropriate software for data processing (following ASTM (2010) standard) [22]. The samples were cut into 80 mm × 10 mm pieces and stretched at a rate of 250.00 mm·min^−1^ to measure tensile strength, elongation at break and maximum applied force at room temperature. Tests were performed for at least five samples of each formulation [23].

### 2.12. Determination of Surface Color

Surface color was measured using a chromometer (Konica Minolta, CR-400, Tokyo, Japan) and expressed as *L** (lightness), *a** (redness/greenness) and *b** (yellowness/blueness) values. Hunter color values (*L*, *a*, and *b*) were determined from the average of five readings at five points in the center of the specimen and at four spread-around points for each sample. A white color plate (*L* = 90.00, *a* = 1.36, *b* = −1.47) was employed as the standard background. The total color difference (*ΔE*) was calculated as:(5)$$\Delta E=\;\sqrt{{\left(\Delta L\right)}^{2}+{\left(\Delta a\right)}^{2}+{\left(\Delta b\right)}^{2}},\;$$
where Δ*L*, Δ*a* and Δ*b* are the differences between the color values of the standard color plate and those of the film samples [24].

### 2.13. Determination of Light Transmittance

The optical properties of composite films were probed by light absorption measurements using a UV-Vis spectrophotometer (Shimadzu UV-1800, Kyoto, Japan) according to the method of Shankar et al. Film strips with dimensions of 10 mm × 60 mm were placed in the spectrophotometer test cell, and an empty test cell was used as a reference. UV barrier properties and transparency were determined from light transmittances at 280 and 660 nm, respectively. For each film, measurements were performed in triplicate, and the average of three spectra was calculated.

### 2.14. Evaluation of Biodegradability

Biodegradability was determined by measuring the *weight loss* of the membranes buried under soil [25]. For the biodegradability analysis, the membranes were cut into 6 cm × 1 cm pieces (further 3 cm × 3 cm samples were taken for photographing), weighed, tied at one corner with a thread and buried about 15 cm below the surface of the soil. The buried membranes were removed from the soil every two weeks, washed by deionized water, and dried at 60 °C until the weight of the films did not change. The *weight loss* was then calculated using the following equation:(6)$$Weight\;loss=\;\frac{W_{i}-W_{d}}{W_{i}},\;$$
where *W_i_* = initial weight of the specimen and *W_d_* = dry weight of the specimen after degradation in soil.

### 2.15. Evaluation of Antimicrobial Activity

Samples from the experimental and control groups were cut into pieces, and 0.5 g (dry weight) specimens were weighed and placed in Erlenmeyer flasks for high-temperature sterilization. Other Erlenmeyer flasks (not containing samples) were filled with 50 mL of NA solutions (Take 500 mL as an example: 1 mL nutrient broth (Take 100 mL as an example: 0.3 g beef extract, 1 g peptone, 0.5 g NaCl, 100 mL distilled water) + 499 mL 0.8% NaCl aq (Take 100 mL as an example: 0.8 g NaCl, 100 mL distilled water)). *Escherichia coli* and *Staphylococcus aureus* were transferred via inoculation loops into flasks filled with 150 mL of nutrient broth and incubated for 24 h at 37 °C with stirring. Subsequently, 1 mL of the incubated broth medium was injected into two Erlenmeyer flasks each, one containing the sample, and the other containing the NA solution. The two flasks were incubated at 37 °C for 24 h with stirring, following which a 100 μL aliquot of the incubated sample solution was suctioned and uniformly coated on a solid medium. This step was followed by 24 h incubation in a constant-temperature incubator at 37 °C. The incubated solid medium was then removed and subjected to colony counting [14].

### 2.16. Preservation Experiment of Strawberries

A batch of fresh strawberries (purchased from Dandong, Liaoning) was transported by cold chain. We selecting strawberries of moderate size and similar ripeness and then packed them with different films. The unpacked strawberries were the blank group, and the strawberries with PE films were the control group. The other four groups were wrapped in PVA/CMC, PVA/CMC/ZnO NPs, PVA/CMC/ZnO NPs/*x*GnP (ZnO NPs:*x*GnP = 7:3) and PVA/CMC/*x*GnP (ultrasonication time, 30 min) films. All the samples were kept at 25 ± 1 °C and 55 ± 5% RH for 6 days. The preservation indexes were detected every 2 days.

#### 2.16.1. Weight Loss Ratio

The weight of strawberries was measured every two days to calculate the *weight loss ratio* by the following Equation (7):(7)$$Weight\;loss\;ratio\;\left(\%\right)=\;\frac{W_{0}-W_{t}}{W_{0}}\;\times 100\%,\;$$
where *W*_0_ is the initial weight of strawberries (g) and *W_t_* is the sample weight after storage time (g).

#### 2.16.2. Firmness

The firmness of the strawberries was measured by a fruit hardness tester (GY-4, Handpi, Zhejiang, China) using a flat 4 mm cylindrical probe, which was pressed into three different points in the central zone of each strawberry to a depth of 10 mm [26].

#### 2.16.3. Content of Total Soluble Solids (TSS)

The TSS in the strawberry pulp was determined by an Abbe Refractometer (2WAJ, Shanghai Optical Instrument Co., Ltd., Shanghai, China) and expressed as a percentage.

#### 2.16.4. Titratable Acidity (TA)

The TA was determined by the titration of 5 mL of juice with 0.1 mol/L NaOH using phenolphthalein as the indicator, and the results were expressed as percent citric acid [27].

### 2.17. Statistical Analysis

The final result was expressed as the mean ± standard deviation. The SPSS 24.0 statistical analysis system was used for analysis of variance (ANOVA) and Duncan’s multi-range tests were used for determining significant differences between the groups (*p* < 0.05).

## 3. Results and Discussion

### 3.1. Chemical Property–Swelling Property

Figure 1 shows the dissolution and swelling condition of PVA/CMC, PVA/CMC/ZnO NPs, PVA/CMC/ZnO NPs/xGnP (ZnO NPs:*x*GnP = 9:1, 7:3, 5:5, 3:7, 1:9), PVA/CMC/*x*GnP film with 30 min ultrasonication and PVA/CMC/ZnO NPs/*x*GnP (ZnO NPs:*x*GnP = 7:3) composite films under different ultrasonication times. During the swelling experiment, all films were wrinkled when they came into contact with water, and the most severe degree of shrinkage was seen with the PVA/CMC film. Is well known that PVA has a high degree of swelling in aqueous solvents [28]. Similarly, CMC also has good hydrophilicity. In addition, we observed that there were holes in the PVA/CMC/*x*GnP film, possibly as a result of the formation of extra pores on the polymer’s surface due to the ultrasonication [29]. This could also be because *x*GnP is prone to agglomeration under the action of van der Waals forces and π–π stacking interactions, which leads to the uneven dispersion of the solution and the appearance of pores.

### 3.2. Physical Property

#### 3.2.1. Composite Morphology

Figure 2A shows that the PVA/CMC film had a clear and uniform surface morphology due to good cohesion between its components. This phenomenon corroborated the findings of Fasihi et al. [30], who suggested that the system has a good miscibility ascribed to the H-bond formed between PVA and CMC [19]. The presence of visible spherical particles in the case of films containing ZnO NPs (Figure 2A) was ascribed to particle agglomeration [31]. Further, with the ratio of ZnO NPs and *x*GnP changing from 0:10 to 10:0, the number of visible spherical particles increased continuously. However, PVA/CMC/*x*GnP films contained aggregates, which was ascribed to the strong attractive interactions between graphene nanosheets preventing their good distribution in the matrix [32]. At the same time, the chemically inert *x*GnP is prone to agglomeration under the action of van der Waals forces and π-π stacking interactions. To obtain a homogenous construction, the nanoparticulate fillers should be separated from each other as much as possible [33]. In contrast, the composite films prepared herein featured a rough texture (Figure 2Ah) in the form of wrinkles and corrugations, which were primarily due to the stacking of graphene sheets [34]. The *x*GnP-containing films featured flake-shaped structures, the number of which increased with increasing *x*GnP content. At a fixed ZnO NPs + *x*GnP content, a ZnO NPs:*x*GnP ratio of 7:3 resulted in the most uniform particle dispersion. Figure 2Ag shows the presence of *x*GnP wrinkles and non-uniform dark flakes. At the same time, the adhesion between the particles and the matrix is quite poor, as revealed by the presence of some voids around the *x*GnP. The cross-sectional SEM images of the PVA/CMC/ZnO NPs/*x*GnP composite films showed the internal reticular structure, which was more fluffy than that of the untreated PVA/CMC film. The presence of ZnO NPs and *x*GnP in the composite nanofibers was further confirmed by EDS analysis (Figure 2B).

#### 3.2.2. ATR-FTIR Analysis

Figure 3 shows the ATR-FTIR spectra of *x*GnP, ZnO NPs, PVA/CMC, PVA/CMC/ZnO NPs, PVA/CMC/*x*GnP and PVA/CMC/ZnO NPs/*x*GnP films. The spectrum of ZnO NPs showed a peak at 1630 cm^−1^, which is attributed to the O–H bonds of water absorbed on the particle surface [35]. *x*GnP showed peaks at 3440 and 1630 cm^−1^ due to the O–H stretching vibration of absorbed water and the C=C skeletal vibration of graphene, respectively [19]. The low-frequency scattering to the right of the latter peak (in the region of 1200–1600 cm^−1^) was related to the buffer layer [36]. The spectrum of the PVA/CMC film showed characteristic signals of PVA and CMC, namely the broad peaks at 3500–3100 cm^−1^ (O–H stretching vibrations in PVA and CMC) and narrow peaks at 2920 and 1060 cm^−1^ (aliphatic C–H and C–O stretches in PVA, respectively) [37]. The peaks at 1060, 1320, 1420 and 1600 cm^−1^ were characteristic of CMC, and corresponded to C–O–C bending, –OH bending, –CH2– scissoring and asymmetric –COO– vibrations, respectively. Samsudin et al. [38] reported similar CMC signature bands at 1056, 1334, 1421 and 1581 cm^−1^, which were carbohydrate signature peaks, confirming the presence of carboxymethyl substituents on the CMC backbone [19]. For the PVA/CMC/ZnO NPs film, the intensities of the above FTIR peaks exceeded those observed for the pure PVA/CMC film. This behavior indicated that some strong inter-component interactions and good dispersions changed the polymer chain arrangement, which was ultimately reflected in the increased peak intensity. The FTIR spectra of PVA/CMC/ZnO NPs films indicated that doping with ZnO NPs influenced the functional groups of PVA/CMC matrices as a result of the complexation or interaction of these groups with ZnO NPs [5]. The FTIR spectra of PVA/CMC/ZnO NPs films revealed a marginal shift in the positions of the bands corresponding to OH and C=O stretches, which was attributed to the interaction between the ZnO NPs filler and the host PVA/CMC matrix (Figure 3) [39]. By comparing the PVA/CMC/ZnO NPs/*x*GnP films and the PVA/CMC/ZnO NPs films, we can see that with the ratio of ZnO NPs and *x*GnP changing from 5:5 to 10:0, the intensity of the –OH peak at 3270 cm^−1^ increased and shifted to higher wavenumbers, while that of the COO− and C–O peaks increased and broadened, which was attributed to the strong interaction between the above groups and ZnO NPs [5]. The spectrum of the PVA/CMC/*x*GnP film showed that upon the addition of *x*GnP to PVA/CMC, the intensity of other peaks related to PVA/CMC decreased, while the signature band of *x*GnP at 3440 cm^−1^ became red-shifted and broadened, possibly because of the synergetic influence of strain and doping [36]. The characteristic peaks of the PVA/CMC/ZnO NPs/*x*GnP films were similar to (but weaker than) those of the PVA/CMC film, and the OH stretch of the former at 3380 cm^−1^ and the asymmetric –COO− stretch at 1600 cm^−1^ red-shifted to 3260 and 1590 cm^−1^, respectively, in case of the latter, which indicated a strong interaction between these groups and ZnO NPs or *x*GnP [25]. The absence of new peaks suggested that the interactions between ZnO NPs or *x*GnP and the PVA/CMC matrix were purely physical (e.g., hydrogen bonds and van der Waals forces).

Ultrasonication did not change the overall appearance of the FTIR spectra, mainly affecting peak intensity and position [13]. Figure 4 shows the FTIR spectra of PVA/CMC/ZnO NPs/*x*GnP composite films prepared using different ultrasonication times. The intensity of the characteristic peak of PVA/CMC decreased after the short ultrasonic treatment, which was ascribed to its promotional effect on the uniform dispersion of ZnO NPs and *x*GnP in the PVA/CMC matrix and the closer incorporation of the nanofillers into this matrix. However, in the case of the long (60 min) ultrasonication treatment, the peak intensity of PVA/CMC slightly increased, possibly because of the release of the intermolecular hydroxyl groups of PVA/CMC.

#### 3.2.3. XRD Analysis

Figure 5 shows the XRD patterns of ZnO NPs, *x*GnP, PVA/CMC, PVA/CMC/ZnO NPs, PVA/CMC/*x*GnP and PVA/CMC/ZnO NPs/*x*GnP films. The significant peaks of ZnO NPs at 2*θ* = 31.75°, 34.37°, 36.26°, 47.56°, 56.59°, 62.85°, 67.96°, 69.05°, 72.55° and 76.94° were ascribed to the reflections from the (100), (002), (101), (102), (110), (103), (112), (201), (004) and (202) planes of hexagonal ZnO NPs with a wurtzite structure (space group P63mc, JCPDS No. 36 1451), respectively. No additional peaks due to secondary or impurity phases were observed, which confirmed the phase purity of the ZnO NPs sample. The pattern of *x*GnP featured an intense peak at 2*θ* = 26.48°, attributed to the stacking of single graphene layers at a distance of 0.34 nm, and other peaks at 2*θ* = 43.96° and 54.62°, corresponding to reflections from the (110) and (102) planes, respectively [40]. In agreement with the findings of Goswami et al. [41] and the known semi-crystalline nature of PVA, as well as the amorphous nature of CMC, the XRD pattern of PVA/CMC displayed broad peaks at 2*θ* = 19.46° and 40.50°, which indicated the existence of a typical semi-crystalline structure and suggested that PVA strongly interacted with CMC [42]. The pattern of the PVA/CMC/ZnO NPs film showed the characteristic peaks of ZnO NPs at 2*θ* = 31.75°, 34.37°, 36.26°, 47.56°, 56.59°, 62.85°, 67.96° and 69.05°, as well as those of the PVA/CMC film at 2*θ* = 19.46° and 40.50°. Thus, the introduction of ZnO NPs did not change the crystal structure of the PVA/CMC matrix, and the ZnO nanostructures were well crystallized in the polymer matrix as no new peaks or peak shifts were observed. The patterns of the PVA/CMC/*x*GnP and PVA/CMC/ZnO NPs/*x*GnP films showed the peak of semi-crystalline PVA at 2*θ* = 19.46°, and the characteristic peaks of *x*GnP at 2*θ* = 26.48° and 54.62°, indicating that *x*GnP sheets could not be dispersed or completely separated, with some sheets existing in a stacked form [43]. Compared with those of the PVA/CMC/*x*GnP film, the peaks of the PVA/CMC/ZnO NPs/*x*GnP film at 2*θ* = 26.48° and 54.62° were significantly weakened, i.e., the intensity of these peaks increased with the increasing loading of *x*GnP, possibly because of the concomitant increase in the number of stacked *x*GnP layers. The pattern of the PVA/CMC/ZnO NPs/*x*GnP film also featured the characteristic peaks of ZnO NPs at 2*θ* = 31.75°, 34.37° and 36.26°, and the decreased intensity of these peaks (compared with that observed for ZnO NPs) implied that the ZnO NPs were incorporated into PVA/CMC [5].

#### 3.2.4. DSC Analysis

The DSC curve of the PVA/CMC film exhibited three endothermic peaks at 92.52, 222.03 and 313.69 °C, as well as a small exothermic peak at 289.65 °C (Figure 6 and Table 1). The peaks at 92.52 and 222.03 °C indicated the presence of two phases, and suggested that the two polymers were partially miscible and engaged in some sort of interaction with each other [41]. Had this not been the case, the transitions of each polymer in the PVA/CMC blend would occur at similar vitreous transition temperatures [41]. The peak at 289.65 °C is considered to be the crystalline transition. The addition of ZnO NPs and *x*GnP increased the *T_g_* of the composite film, with values of 115.48, 119.92 and 99.07 °C observed for PVA/CMC/ZnO NPs, PVA/CMC/*x*GnP and PVA/CMC/ZnO NPs/*x*GnP, respectively. This behavior was ascribed to the good compatibility between ZnO NPs/*x*GnP and the PVA/CMC polymer matrix, as well as to the strong interfacial interactions between these components, which reduced the mobility of chain segments and allowed chain segment motion to occur only at higher temperatures [44].

Melt characteristics are important parameters for studying composite thermal stability, as they can reflect intermolecular bonding strength in polymer blends [45]. The melting process can be regarded largely as unaffected by ZnO or graphene. Compared to those of the PVA/CMC film (*T_m_* = 221.80 °C), the Tm values of the PVA/CMC/*x*GnP films were slightly lower (220.24 °C), possibly because the good dispersion of nanofillers in the PVA/CMC matrix improved component compatibility and allowed the formation of exfoliated and/or embedded composite films [32]. The *T_g_* value of the pure PVA/CMC film was 92.52 °C, which is in line with that reported in the literature [46]. Compared to those of the PVA/CMC films, the *T_g_* values of the composite films with added ZnO NPs or *x*GnP were higher. For the rest of the composites, the nanoparticles can induce an increase in the *T_g_*. The thermal stability of nanocomposites depends on the state of the filler dispersion, interfacial interactions between the filler and the matrix, filler particle size, and polymer molecular weight and crystallinity [47]. The enhanced crystallinity of the composite film observed herein was ascribed to the nucleation of *x*GnP and ZnO NPs. Figure 5 reveals that a strong characteristic peak at 26.48° was observed for *x*GnP, indicating that *x*GnP itself was highly ordered and thus implying that a large number of polymer chains could possibly be effectively aligned along the crystal nuclei (*x*GnP), in order to induce crystallization and improve crystallinity [48]. This suggestion is in line with the reported ability of reinforcing substances to serve as nucleating agents and thus provide a surface for the heterogeneous crystallization of polymers [47].

#### 3.2.5. WVP

Figure 7 shows that with the decreasing ZnO NPs:*x*GnP ratio, the WVP of composite films first decreased and then increased, lying in the range of 3.92–9.88 × 10^−13^ g·Pa^−1^·s^−1^·cm^−1^. The reduction in WVP upon the introduction of ZnO NPs, reported for many bionanocomposites, is mainly ascribed to the hydrophobic nature of this nanofiller and the concomitant generation of a tortuous pathway for water vapor molecules to pass through [24]. The minimum WVP of 5.29 ± 0.13 × 10^−13^ g·Pa^−1^·s^−1^ cm^−1^ (observed at a ZnO NPs:*x*GnP ratio of 5:5) was 35.55% lower than that of the pure PVA/CMC film, which was attributed to the change in the original molecular chain structure of PVA/CMC and the interfacial interaction between ZnO NPs NPs and *x*GnP due to the binding of ZnO NPs NPs with *x*GnP. In contrast to that of ZnO NPs, the graphene–PVA/CMC binding force was believed to be weak, resulting in increased WVP because of the formation of numerous pores on the surface of the composite film upon *x*GnP addition. In a word, both ZnO NPs and *x*GnP had a profound impact on WVP, which increased as a result of crystallinity reduction and decreased because of the formation of a tortuous path due to the nucleation between ZnO NPs and *x*GnP, in line with DSC results (Figure 6).

With increasing ultrasonication time, the WVP of composite films first increased and then decreased. The WVP increase was ascribed to the thermal effect of ultrasonication, which enhanced molecular movement and exposed the hydrophilic groups of PVA molecules. The highest WVP of the PVA/CMC film (9.67 ± 0.21 × 10^−13^ g·Pa^−1^·s^−1^·cm^−1^) was obtained at an ultrasonication time of 30 min. The strong thermal, mechanical, super-mixing and cavitation effects of ultrasound treatment can make the particles oscillate and collide at high speed to break the hydrogen and hydrophobic bonds in the medium, thus facilitating the formation of nonpolar bonds and, hence, the process of film densification [12]. Moreover, ultrasonication further improves material solubility, makes the film form a more compact network structure, and induces stirring that promotes the uniform macro-scale distribution of particles in solution. The cavitation effect can further homogenize particles on the micro scale, facilitate the separation and uniform distribution of graphene and ZnO NPs in the matrix, and increase the compactness of the composite film to reduce WVP [14]. Notably, the PVA/CMC film subjected to 60 min ultrasonication did not feature the lowest WVP, possibly because this time was insufficient for polymer destruction.

#### 3.2.6. Mechanical Properties

Figure 8a shows that the addition of ZnO NPs and *x*GnP significantly changed the elongation at break (*p* < 0.05), which first increased and then decreased with decreasing ZnO NPs:*x*GnP ratio. At a ZnO NPs:*x*GnP ratio of 10:0, the composite film exhibited a strain of 164.56 ± 6.64%, which was 22.77% higher than that of the pure PVA/CMC film (134.04 ± 7.43%). This behavior was ascribed to the stiffness of hter ZnO NPs and their favorable hydrogen bonding interactions with PVA [49]. The lowest elongation at break was observed at a ZnO NPs:*x*GnP ratio of 5:5, possibly because the fillers themselves and especially the filler–matrix interface are stressed, and may lose structural integrity, which may manifest as the formation of holes in the matrix that create an initial fracture [48]. The increase in elongation at break initially observed with increasing *x*GnP dosage was attributed to the good dispersion of *x*GnP achieved during compounding [50]. This behavior was ascribed to the strong adsorption of graphene, which may limit the movement of PVA/CMC molecular chains to increase tensile strength. Moreover, graphene can increase compatibility between ZnO NPs and PVA/CMC and provide the effect of particle reinforcement [51]. Finally, the beneficial effect of *x*GnP can also be ascribed to the high mechanical strength of graphene sheets [52]. As the graphene loading increased from 0 to 10, the elongation at break increased from 134.04 ± 7.43% to 175.61 ± 7.38%, decreasing at higher loadings. This behavior was rationalized as follows. An increase in graphene loading induces the internal stratification of the composite, and the accumulation of graphene sheets leads to stress concentration and, hence, to premature failure and reduced elongation at break.

The tensile strength of a material reflects its resistance to the breaking process when a constant load is applied on the material. Another significant mechanical property is the Young’s modulus, also called the elastic modulus, which is a property of linear elastic solid materials. This property describes the relationship between the stress and the strain of a material [41]. Figure 8c shows that the addition of ZnO NPs and *x*GnP significantly affected the Young’s modulus. This combines with Figure 8b, showing that the variation trend in the tensile strength of the composite film was basically the same as the Young’s modulus (*p* < 0.05). Pure PVA/CMC films featured a maximum tensile strength of 45.50 ± 3.18 MPa and a maximum Young’s modulus of 33.95 ± 2.93 MPa, which was hardly affected by the incorporation of only ZnO NPs or graphene. This finding was ascribed to the poor compatibility between ZnO NPs or *x*GnP and PVC/CMC, which resulted in weak interfacial adhesion between the two phases, and hence in the easy initiation and propagation of cracking at the corresponding interface [51]. However, this impact was small. At a ZnO NPs:*x*GnP ratio of 9:1, the Young’s modulus of the composite film sharply decreased to 2.61 ± 0.62 MPa, which was 92.14% lower than that of the pure PVA/CMC film (33.21 ± 2.90 MPa), i.e., the addition of *x*GnP reduced the plasticity of the material. This behavior was in line with the results of Ranjan et al. [53], and was ascribed to the antagonism between crack bridging, crack arrest and strong interface formation.

With increasing the ultrasonication time, the elongation at break of the composite films first increased and then decreased, which was explained as follows. The intense vibration induced by ultrasonication reduced the matrix porosity and the matrix–fiber clearance to increase the tensile strength and afford more compact structures, as well as improving the mechanical properties and allowing good film formation, as reported previously [54,55]. Moreover, prolonged ultrasonication led to the formation and exposure of more reaction centers, accelerating the rate of chemical reactions and allowing the faster recombination of macromolecules in order to increase tensile strength. The concomitant formation of a more closely bound network structure resulted in increased elongation at break. The mechanical properties of pure PVA/CMC films improved with increasing ultrasonication time, and were optimal (2.26% higher elongation at break than that of the pure PVA/CMC film) in the case of the 15 min ultrasonication. However, the elongation at break decreased for longer treatment times, as the excessive polymer damage inflicted under these conditions resulted in the formation of extra pores on the polymer surface, and hence reduced density [54]. At an ultrasonication time of 60 min, the elongation at break was 21.71% lower than that observed for the non-ultrasonicated film.

#### 3.2.7. Surface Color

Figure 9 shows the pictures of the pure PVA/CMC film and the PVA/CMC/ZnO NPs/*x*GnP composite film (ultrasonication time, 30 min), corresponding to Table A1. Table A1 lists the colors and transmittances of the PVA/CMC and PVA/CMC/ZnO NPs/*x*GnP films, revealing that the PVA/CMC film was colorless and transparent, while the PVA/CMC/ZnO NPs film was white and featured a higher *L* value than the former film (*p* > 0.05), which was ascribed to the whitening effect of ZnO NPs [24]. The incorporation of *x*GnP into the PVA/CMC/ZnO NPs/*x*GnP films resulted in darker colors, significantly decreasing *L* and increasing *a* and *b* (*p* < 0.05), and the extent of these changes was found to be elevated with increasing *x*GnP content. At the lowest *x*GnP content, the *L* value (48.70 ± 1.12) was 46.35% lower than that of the PVA/CMC film (90.78 ± 0.26), and the *ΔE* value was significantly higher (42.35 ± 0.95) than that of the PVA/CMC film (1.13 ± 0.04), i.e., *x*GnP had a significant effect on film color. This result was in line with the findings of Zhang et al. [25], who demonstrated that the incorporation of graphene oxide into PVA-based composite films resulted in their darkening (*p* > 0.05).

Ultrasonic treatment also influenced film color and transmittance. Table A1 shows that ultrasonication did not significantly change the *L**, *a** or *b** values of PVA/CMC, PVA/CMC/ZnO NPs and PVA/CMC/*x*GnP films (*p* > 0.05), but significantly affected the *L** value of some PVA/CMC/ZnO NPs/*x*GnP films. This behavior was explained by the beneficial effects of heating and cavitation, generated by ultrasonic treatment, on the dispersion of *x*GnP in water at low *x*GnP content and high ZnO NPs content [56]. Thus, under these conditions, *x*GnP was uniformly distributed in the composite film (because of their high content, ZnO NPs easily agglomerated in the PVA/CMC matrix) to result in a dark color. In addition, proper ultrasonication (15, 30 and 45 min) could reduce the transmittance of the composite film (*p* > 0.05). However, when the ultrasonication time increased to 60 min, the transmittance of the composite film increased (*p* < 0.05), possibly because overly long ultrasonic treatment reduced the viscosity of the film-forming solution and polymer molecular weight by inflicting damage on the network structure of PVA/CMC, thus increasing the amount of holes on the composite film’s surface [57,58].

#### 3.2.8. Antimicrobial Activity

Figure 10a shows that whereas PVA/CMC films did not show any antimicrobial activity, films doped with ZnO NPs and *x*GnP had good antibacterial activity, which was positively correlated with ZnO NPs content. The highest antibacterial activity was observed for the PVA/CMC/ZnO NPs film, which inhibited the proliferation of *E. coli* and *S. aureus* by 94.53 ± 1.85% and 96.42 ± 2.32%, respectively. Composite films containing ZnO NPs had a stronger inhibitory effect on *S. aureus* than on *E. coli*, which was related to the mechanism of ZnO NPs action and the cell structure of the two bacteria. According to Espiti et al. [8], Gram-negative bacteria have an additional outer film compared with Gram-positive bacteria, and the negative charge of the film of *S. aureus* is smaller than that of *E. coli*. Hence, *S. aureus* more strongly interacts with the negatively charged reactive oxygen species generated by ZnO NPs (which, however, is only one mechanism of its antibacterial action).

With increasing *x*GnP content, antibacterial activity weakened, and the inhibition efficiencies of the PVA/CMC/*x*GnP film for *E. coli* and *S. aureus* decreased to 48.30 ± 2.31% and 50.02 ± 2.74%, respectively, which values were 48.91% and 48.12% lower than the respective values of the PVA/CMC/ZnO NPs film. This behavior was ascribed to the fact that *x*GnP has a weaker antibacterial effect than ZnO NPs [50]. At present, the antibacterial activity of *x*GnP is mainly rationalized as follows: (i) The sharp edges of *x*GnP can physically damage bacterial cell films, inducing causing the leakage of intracellular materials and, thus, cell death; (ii) *x*GnP internalized by bacterial endocytosis and phagocytosis can promote the generation of reactive oxygen species in bacterial cells, thereby inactivating bacteria; (iii) Given that the bacterial cell surface is negatively charged, *x*GnP can act as a “bridge” for charge transfer to destroy film integrity and cause bacterial death; (iv) As *x*GnP has a flexible plane and a strong adsorption capacity, it can be adsorbed by bacterial cells to inhibit their proliferation and even induce cell death [59]. However, studies on the antibacterial activity of *x*GnP are few in number. Girdthep et al. [48] reported that the good interaction between E. coli and hydrophobic *x*GnP favors the attachment of these bacterial cells to *x*GnP, which causes cell lysis. Akhavan et al. [60] also found that upon contact with bacterial cells, *x*GnP can severely damage their films and inactivate these cells.

With increasing ultrasonication time, the antibacterial activity of composite films increased, but the difference between treatment times was not significant (*p* > 0.05) (Figure 10b). After 60 min ultrasonication, the *E. coli* and *S. aureus* inhibition efficiencies increased to 92.04 ± 2.00% and 93.70 ± 3.91%, respectively, which values were 1.83% and 1.49% higher than the values observed without ultrasonication, respectively, in agreement with the results previously reported by Zhang et al. [25]. This behavior was ascribed to nanofiller dispersion into smaller aggregates upon ultrasonication, which increased the specific surface area of ZnO NPs and *x*GnP, increasing the probability of contact between the nanofiller and microbial cells, thereby enhancing the antibacterial effect. In addition, the release of kinetic and thermal energy upon ultrasonication facilitated the release of ZnO NPs and *x*GnP in the polymer, improving antibacterial activity [14]. Therefore, the use of ultrasonic treatment to improve the antibacterial properties of composite films is worthy of further research.

### 3.3. Safety Issues

We consulted a series of related documents in the early stages of the experiment and found that the migration number of nanomaterials was within the scope of safety standards. For example, Heydari-Majd et al. [61] studied the content of Zn^2+^ ions in fish fillets wrapped in PLA/ZnO composite membranes, and found that the migration quantities of Zn^2+^ ions from the nanocomposite membrane to the fillet measured up to 1.551 ± 0.160 mg/100 g sample, which was still far below the migration limit of 40 mg/day for zinc daily consumption as defined by the National Institute of Health for food contact materials. Panea et al. [62] analyzed the migration of ZnO and Ag particles in the aqueous food simulant by inductively coupled plasma mass spectrometry (ICP-MS), and found that the migration of nanoparticles in the simulant was very low, the migration of Zn^2+^ in the control package was below the detection limit (<0.005 mg/kg), and in the packaging of the added nanoparticles, only 2.44 ± 0.37 mg/kg of Zn^2+^ concentration was detected, which is well below the limit established by COMMISSION REGULATION (EU) No 10/2011 (25 mg/kg food or food simulant).

### 3.4. Biodegradability

The degradation performances of the PVA/CMC and PVA/CMC/ZnO NPs/*x*GnP composite films with different ZnO NPs:*x*GnP ratios in a natural environment are presented in Table 2. After degradation in soil, these films absorbed water, became sticky and soft, and decreased in size. After 20-day degradation in soil, the weight of all samples significantly decreased (*p* < 0.05) because of the presence of water and microorganisms in the soil, and the degradation efficiencies of composite films were lower than those of the PVA/CMC films [45]. Specifically, under these conditions, degradation efficiencies of 32.49 ± 3.25% and 30.84 ± 2.87% were obtained for the PVA/CMC/ZnO NPs film and the PVA/CMC/ZnO NPs/*x*GnP film with a ZnO NPs:*x*GnP mass ratio of 9:1, respectively, this being only 5.47% and 7.12% lower than that of the PVA/CMC film (37.96 ± 3.60%), respectively. The higher degradability of the PVA/CMC film was ascribed to the hydrophilicity of its components, PVA and CMC, and the easy penetration of soil moisture into the polymer network, which accelerated water absorption and swelling, and weakened interactions between the materials [25]. The elevated water content promoted the growth of microorganisms during the degradation process and thus made the film susceptible to hydrolysis by soil microorganisms, thereby increasing film weight loss [25]. The slight decrease in degradability observed upon the introduction of ZnO NPs and *x*GnP was ascribed to the formation of a denser film structure as a result of the incorporation of these nanofillers between the *x*GnP of the PVA/CMC matrix, which decreased water permeability and reduced the swelling degree, thus decreasing the extent of hydrolysis and microbial attack [63]. Further observations revealed that the degradation efficiency increased with increasing ZnO NPs content, possibly because the content of super-hydrophobic *x*GnP was relatively low, and hydrophilic ZnO NPs particles containing hydroxyl groups on their surface promoted water absorption-induced swelling, thus facilitating degradation. In addition, as ZnO NPs particles easily formed agglomerates and were unevenly dispersed in the PVA/CMC matrix in high concentrations, they formed a weak area in the composite film, reducing the mechanical strength and accelerating water absorption and swelling to promote degradation [64]. This finding was consistent with the results of Lani et al. [65], who showed that the addition of dispersed phases led to a decrease in polymer biodegradation rate. As shown in Table 2, the degradation efficiency of PVA/CMC and composite films increased with increasing ultrasonication time. The highest values were observed for a treatment time of 60 min, equaling 41.54 ± 1.46% for the PVA/CMC film, 37.72 ± 1.36% for the PVA/CMC/ZnO NPs composite film, and 34.27 ± 1.22% for the PVA/CMC/ZnO NPs/*x*GnP composite film with a ZnO NPs:*x*GnP ratio of 9:1. The second and third values were 5.23% and 3.43% higher than those obtained in the absence of ultrasonic treatment, respectively. The promotional effect of ultrasonication on film degradation was ascribed to the influence of ultrasonication-induced heating and cavitation on the polymer chains of PVA and CMC, which was believed to promote the dispersion of nanofillers, hinder their agglomeration, and thus accelerate nanofiller disintegration and release into soil, facilitating the water absorption-induced swelling of the composite film [29].

### 3.5. Preservation Experiment of Strawberries

#### 3.5.1. Photographs of Strawberries During the Storage Time

Figure 11 are the strawberry photographs of typical films during storage time at 25 ± 1 °C, 55 ± 5% RH. During the preservation of strawberries, the PVA/CMC/ZnO NPs/*x*GnP film (7:3) showed the best performance out of all blended films during storage, as the appearance of the strawberries changed minimally compared with other groups. On the sixth day, the strawberry wrapped in a PVA/CMC/ZnO NPs/*x*GnP film had no microorganism growth, while the unpacked group had undergone severe decomposition, and the other groups also had varying degrees of mold growth.

#### 3.5.2. Strawberry Properties

Weight loss is one of the important indexes to judge the freshness of a strawberry. As shown in Table 3, with the extension of storage time, the weightlessness rate of strawberries also showed an increasing trend. During the storage process of strawberries, the body will undergo respiration and evaporation, so that its own water will be lost. The unpacked groups exposed to the environment directly experienced rapid moisture loss, and the poor barrier properties of PE films could allow gasses such as oxygen from the air to permeate, boosting the growth and reproduction of microorganisms on the strawberries and bringing about the rapid decay of fresh fruits. Over the same storage time, the weight loss rates of the unpacked group and the PE film group were relatively high, which were 37.67 ± 1.54% and 35.62 ± 1.68%, respectively. In the other groups of blend films, the strawberries wrapped in PVA/CMC/ZnO NPs/*x*GnP (7:3) films underwent the least weight loss. On the sixth day, their weight loss rate was 18.66 ± 1.99%, 19.01% lower than the unpacked groups. This may be because the composite films on the strawberry surface acted as a semipermeable barrier against gas and water, explaining why PVA/CMC/ZnO NPs/*x*GnP (7:3) films can effectively control the strawberry’s metabolism and water loss.

The hardness of a strawberry is determined by the pectin content in the pulp. The hardness not only affects the taste, but also directly affects the transportation and processing cost. Table 3 shows the trend of strawberry hardness with time in different treatment groups. Starting from the second day, the hardness values of strawberries in each treatment group were significantly different, and the hardness of the unpacked group decreased the fastest, while the hardness of the PVA/CMC/ZnO NPs/*x*GnP film group decreased significantly less than that of the unpacked group. On the sixth day, the hardness of the PVA/CMC/ZnO NPs/*x*GnP film group was 0.41071 ± 0.17647, 0.41071 higher than that of the unpacked group. Moreover, the unpacked group and the PE film group always presented lower firmness values compared with the composite film groups, as strawberries soften by decay.

TSS has a close relationship with fruit maturity and respiration rate, which is one of the evaluation indexes of the strawberry preservation effect. The soluble solids contents of strawberries in different treatment groups during the whole storage period are shown in Table 3. During the first 6 days of the storage period, the content of soluble solids decreased rapidly (down 63.7%) in the unpacked group, which may be because after the post-ripening stage of the fruit, the metabolic rate accelerated and the consumption of soluble solids accelerated. The content of soluble solids decreased slowly (down 33.3%) in the PVA/CMC/ZnO NPs/*x*GnP film group. During the whole storage, the content of soluble solids remained high in the composite film group, significantly higher than that in the unpacked group and the PE film group (*p* < 0.05). This shows that strawberries can have inhibited metabolisms and reduced consumptions of soluble solids after compound coating.

The titration acid content affects the flavor quality of strawberry fruit, which is recognized as one of the important indicators for evaluating the effect of strawberry preservation. Its main component is organic acid. The titration acid content of the composite film group and the unpacked group was not obvious at the beginning. However, from the second day, the change in the titration acid content in each group began to be significant, especially in the unpacked and PE groups. On the sixth day, the titration acid contents of the unpacked group and PE group were 0.31 ± 0.08% and 0.33 ± 0.10%, which were much lower than that of the PVA/CMC/ZnO NPs/*x*GnP film group (0.53 ± 0.06%). This indicates that the breathing and metabolic rate were fast in this period. In addition, the ZnO NPs and *x*GnP film groups had little effect on the titration acid content in the strawberries.

## 4. Conclusions

The target of our work was to design a novel film for food packaging. In our work, PVA/CMC/ZnO NPs/*x*GnP composite films were successfully fabricated by ultrasonication and solution casting, and showed the best overall performance (e.g., strong antibacterial activity against both Gram-positive and Gram-negative foodborne pathogenic bacteria). The best preservation effects for strawberries were as follows: the strawberries wrapped in PVA/CMC/ZnO NPs/*x*GnP (7:3) films showed the least weight loss (19.01% less than the unpacked groups on the sixth day), which was observed at a ZnO NPs:*x*GnP mass ratio of 7:3. In addition, the PVA/CMC/ZnO NPs/*x*GnP composite film (ZnO NPs:*x*GnP mass ratio of 7:3) had a low water vapor transmission rate. By studying the effects of different ultrasonic times on the properties of composite membranes, we found that ultrasonication improved the mechanical properties of composite films and, in combination with traditional film preparation methods and at an ultrasonication time of 30 min, the pure PVA/CMC film exhibited a strain of 165.60 ± 8.88%, which was 31.57% higher than that observed for the non-ultrasonicated film. At the same time, ultrasonic treatment also significantly improves biodegradability. In a word, it was concluded to be well suited for the production of composite films with enhanced mechanical, antibacterial, biodegradability and barrier properties for commercial food packaging applications.

## Figures and Tables

**Figure 1 nanomaterials-10-01797-f001:**
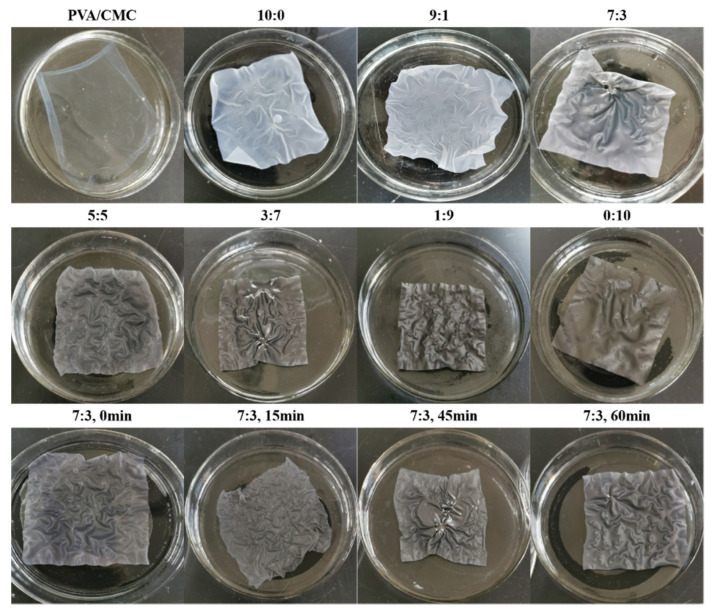
The dissolution and swelling conditions of PVA/CMC, PVA/CMC/ZnO NPs, PVA/CMC/ZnO NPs/xGnP (ZnO NPs:xGnP = 9:1, 7:3, 5:5, 3:7, 1:9), PVA/CMC/*x*GnP film with 30 min ultrasonication and PVA/CMC/ZnO NPs/xGnP (ZnO NPs:*x*GnP = 7:3) composite films under different ultrasonication times.

**Figure 2 nanomaterials-10-01797-f002:**
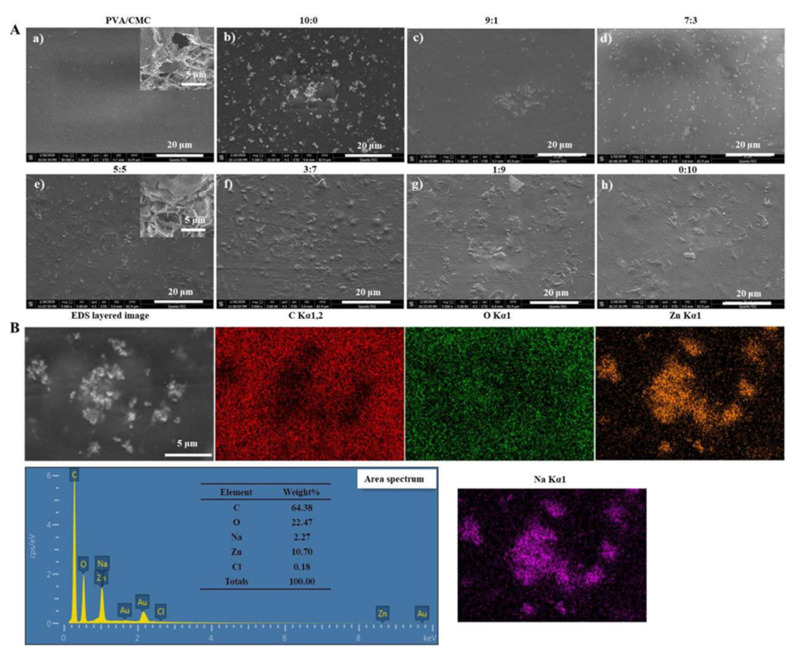
(**A**) SEM (Scanning Electron Microscope) images of (**a**) PVA/CMC film, (**b**–**h**) and PVA/CMC/ZnO NPs/*x*GnP composite films (with different mass ratios of ZnO NPs-to-*x*GnP; ultrasonication time, 30 min). Inset shows cross-sectional SEM images of the PVA/CMC/ZnO NPs/*x*GnP composite film and PVA/CMC film; (**B**) EDS images of PVA/CMC/ZnO NPs/*x*GnP (ZnO NPs:*x*GnP = 7:3) composite film (ultrasonication time, 30 min).

**Figure 3 nanomaterials-10-01797-f003:**
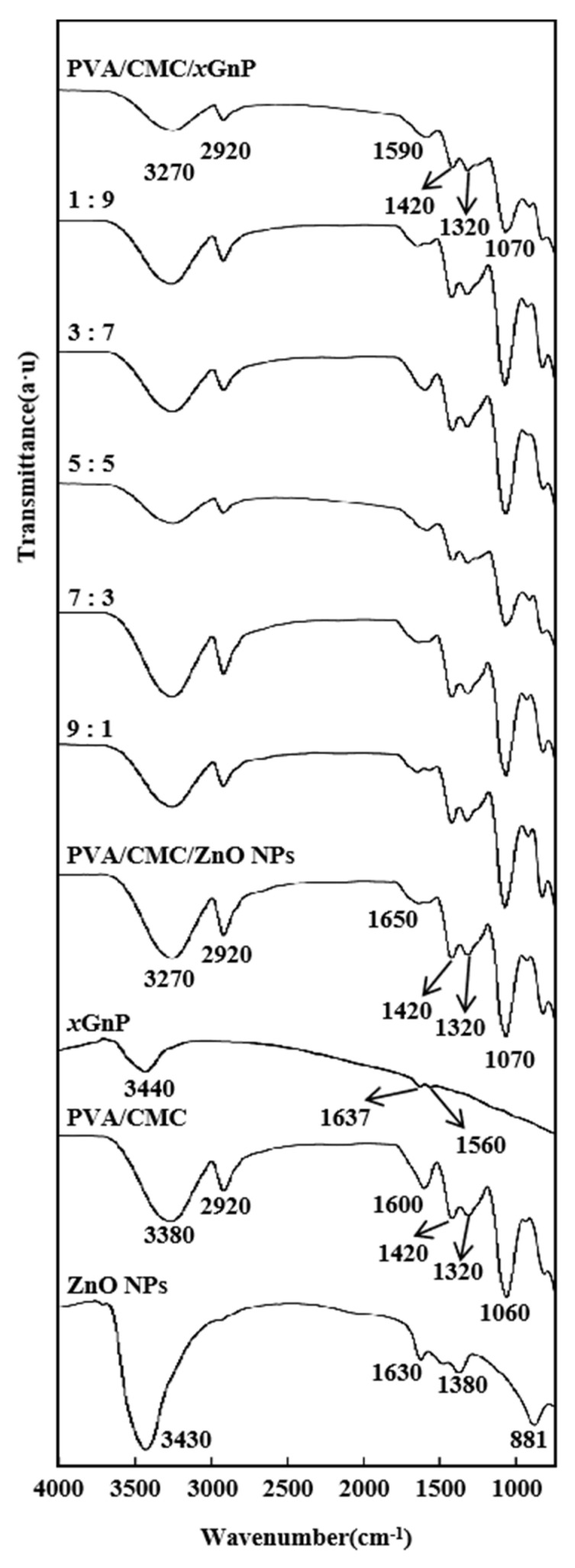
FTIR (Fourier Transform Infrared) spectra of ZnO NPs, PVA/CMC, *x*GnP, PVA/CMC/ZnO NPs, PVA/CMC/ZnO NPs/*x*GnP (ZnO NPs:*x*GnP = 9:1, 7:3, 5:5, 3:7, 1:9) and PVA/CMC/*x*GnP film without ultrasonication.

**Figure 4 nanomaterials-10-01797-f004:**
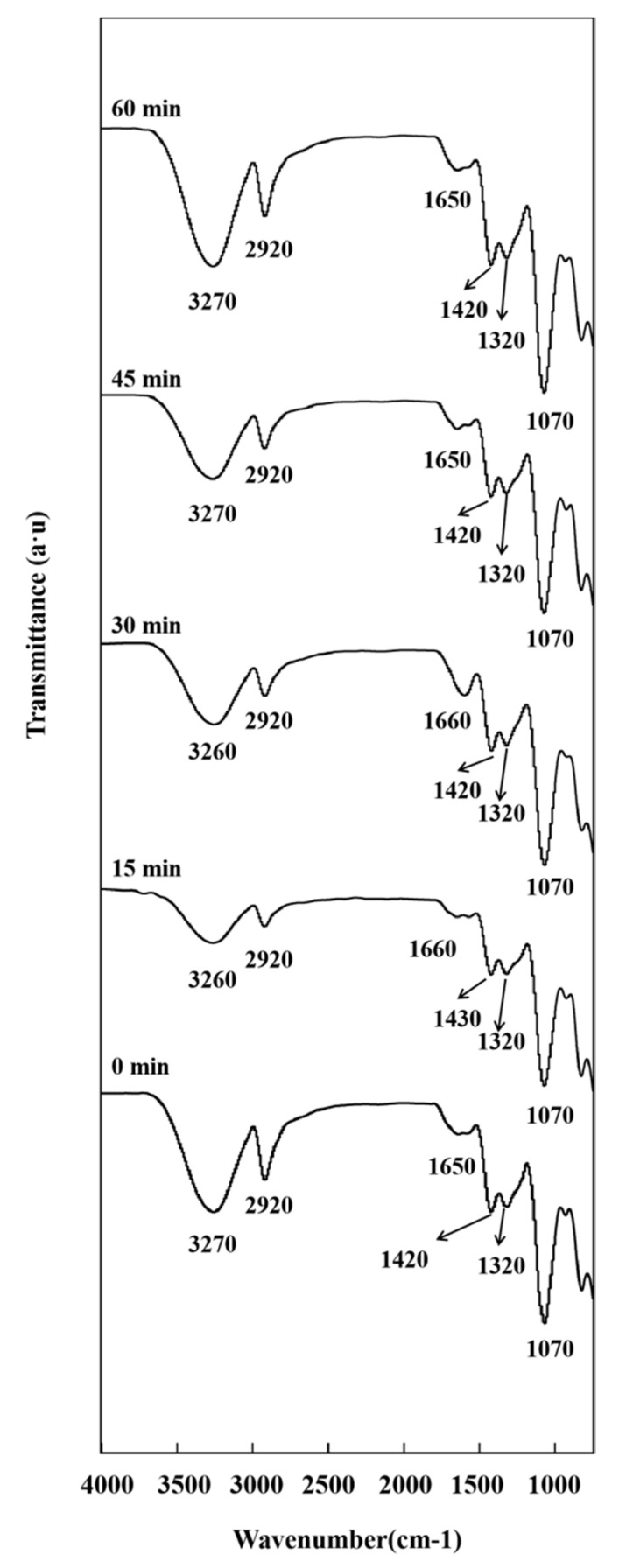
FTIR spectra of PVA/CMC/ZnO NPs/*x*GnP (ZnO NPs:*x*GnP = 5:5) composite films under different ultrasonication times.

**Figure 5 nanomaterials-10-01797-f005:**
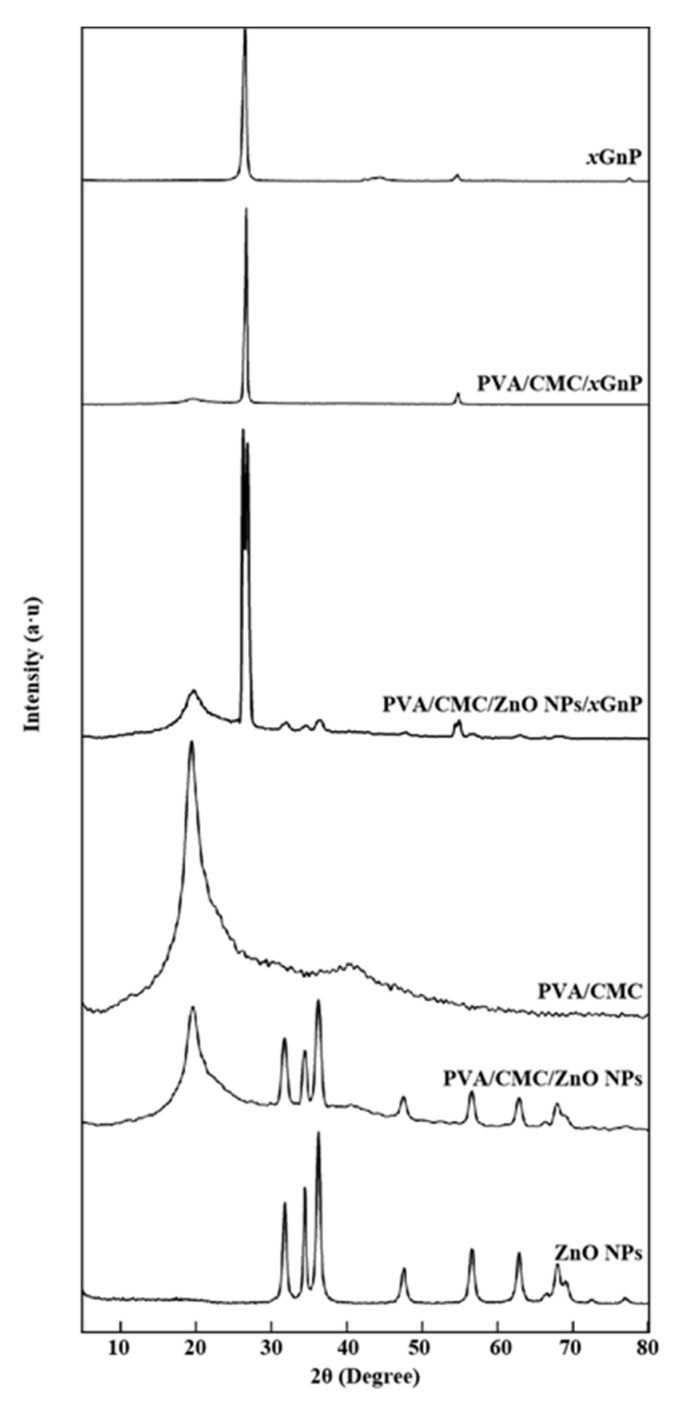
XRD (X-Ray Diffractometry) patterns of ZnO NPs, *x*GnP, the pure PVA/CMC film, PVA/CMC/ZnO NPs, PVA/CMC/*x*GnP, and the PVA/CMC/ZnO NPs/*x*GnP (ZnO NPs:*x*GnP = 7:3) composite film (ultrasonication time, 30 min).

**Figure 6 nanomaterials-10-01797-f006:**
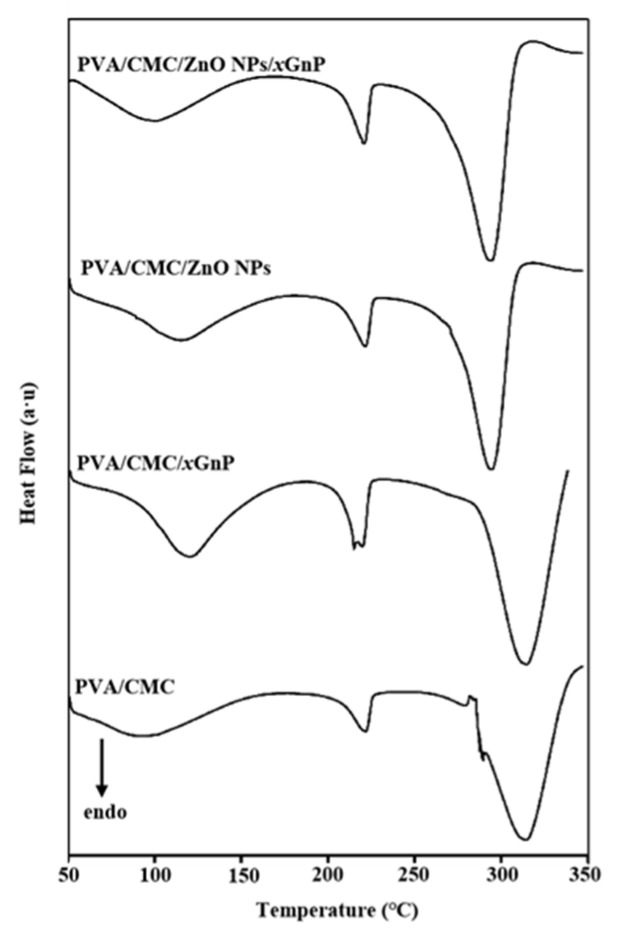
DSC (Differential Scanning Calorimetry) thermograms of pure PVA/CMC, PVA/CMC/*x*GnP, PVA/CMC/ZnO NPs and PVA/CMC/ZnO NPs/*x*GnP (ZnO NPs:*x*GnP = 7:3) composite films without ultrasonication.

**Figure 7 nanomaterials-10-01797-f007:**
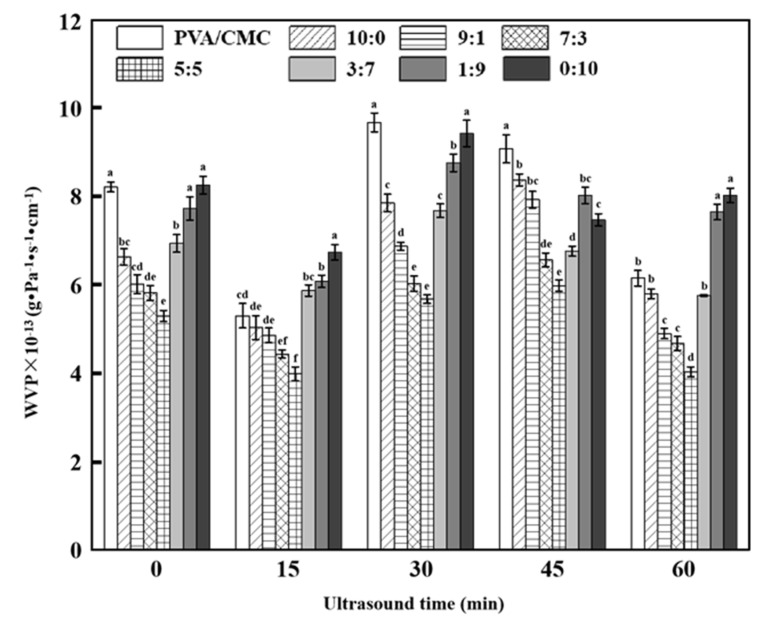
WVP (Determination of Water Vapor Permeability) of pure PVA/CMC and PVA/CMC/ZnO NPs/*x*GnP composite films under different ultrasonication times. (* *p* < 0.05, *n* = 8).

**Figure 8 nanomaterials-10-01797-f008:**
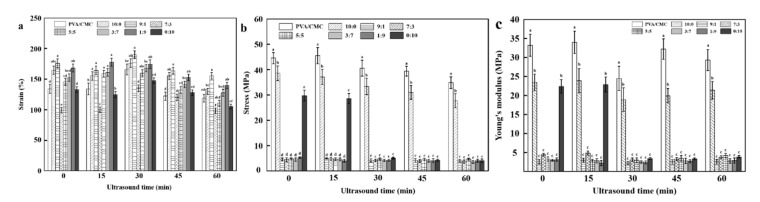
Strain (**a**), Stress (**b**) and Young’s modulus (**c**) of pure PVA/CMC film and PVA/CMC/ZnO NPs/*x*GnP composite films under different ultrasonication times. (* *p* < 0.05, *n* = 8).

**Figure 9 nanomaterials-10-01797-f009:**
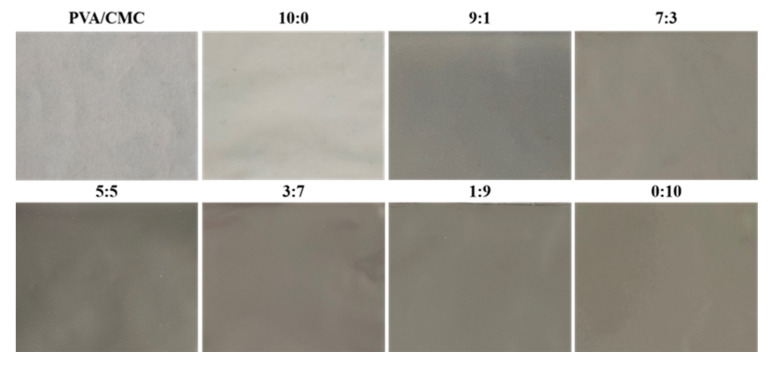
Pictures of the pure PVA/CMC film and the PVA/CMC/ZnO NPs/*x*GnP composite film (ultrasonication time, 30 min).

**Figure 10 nanomaterials-10-01797-f010:**
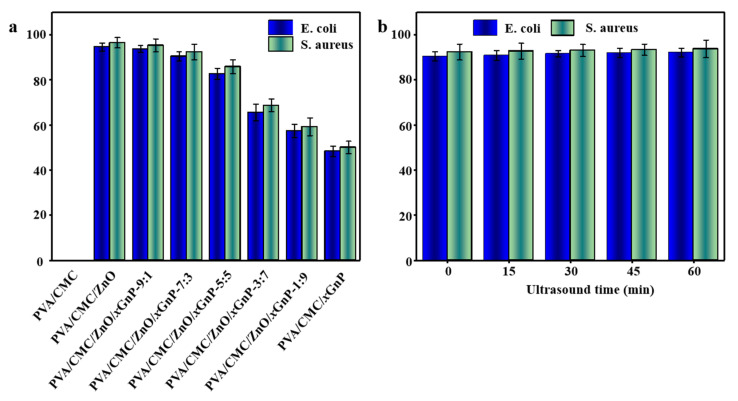
(**a**) The antibacterial activities of the PVA/CMC and PVA/CMC/ZnO NPs/*x*GnP composite films against Gram-positive (*S. aureus*) and Gram-negative (*E. coli*) foodborne pathogenic bacteria; (**b**) The antibacterial activities of the PVA/CMC/ZnO NPs/*x*GnP composite film with a ZnO NPs-to-*x*GnP mass ratio of 7: 3 under different ultrasound times.

**Figure 11 nanomaterials-10-01797-f011:**
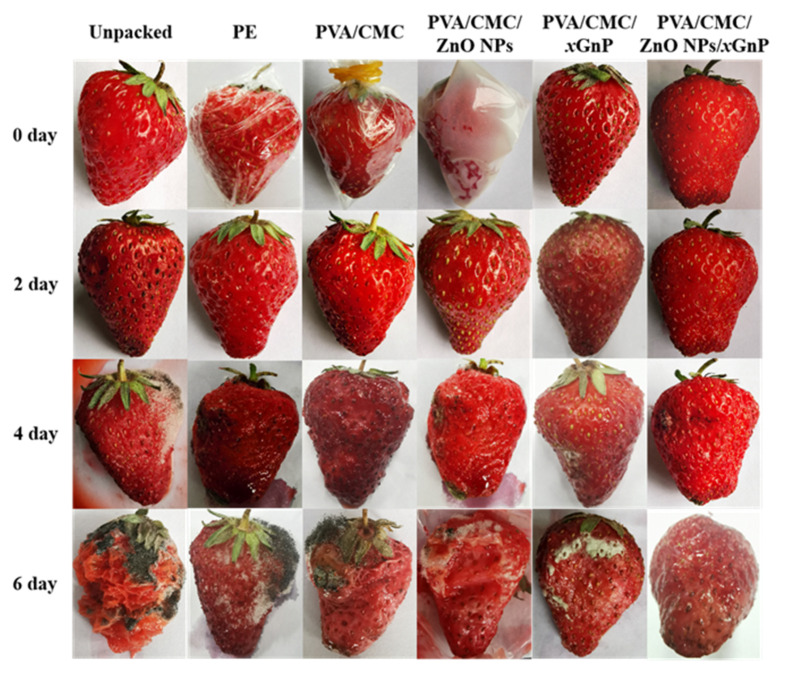
Strawberry photographs of typical films during storage time at 25 ± 1 °C, 55 ± 5% RH.

**Table 1 nanomaterials-10-01797-t001:** The values of *T_g_* and *T_m_* of pure PVA/CMC, PVA/CMC/*x*GnP, PVA/CMC/ZnO NPs and PVA/CMC/ZnO NPs/*x*GnP (ZnO NPs:*x*GnP = 7:3) composite films without ultrasonication.

Sample	*T_g_* (°C)	*T_c_* (°C)	*T_m_* (°C)	*X_c_* (°C)
PVA/CMC	92.52	226.49	221.80	45.52
PVA/CMC/ZnO NPs	115.48	227.48	221.92	49.33
PVA/CMC/*x*GnP	119.92	226.21	220.24	46.94
PVA/CMC/ZnO NPs/*x*GnP	99.07	226.68	221.21	53.20

*T_g_*: transition temperature of the plate glass, *T_c_*: critical temperature, *T_m_*: melting temperature, *X_c_*: capacitive reactance.

**Table 2 nanomaterials-10-01797-t002:** Biodegradation rates of the PVA/CMC film and PVA/CMC/ZnO NPs/*x*GnP composite films under different ultrasonication times. (* *p* < 0.05, *n* = 8).

Time (Days)	Ultrasound Time (min)	Samples							
		PVA/CMC	10: 0	9:1	7:3	5:5	3:7	1:9	0:10
5	0	14.82 ± 1.73Ac	8.72 ± 1.02Bb	8.01 ± 1.21BCb	6.23 ± 0.63BCDb	6.11 ± 0.82BCDb	4.63 ± 0.73CDc	4.22 ± 0.38Dc	4.02 ± 1.32Dc
	15	15.74 ± 1.37Abc	9.58 ± 2.34Bb	9.64 ± 2.09Bb	7.24 ± 1.13Bb	7.08 ± 0.91Bb	5.88 ± 0.56Bbc	4.90 ± 0.44Bc	5.57 ± 0.78Bbc
	30	18.83 ± 2.11Aabc	11.77 ± 1.78Bb	11.56 ± 1.64Bab	9.69 ± 1.57BCab	8.33 ± 0.75BCb	7.49 ± 0.85BCabc	6.46 ± 0.57Cbc	7.72 ± 1.36BCbc
	45	21.54 ± 1.57Aab	14.90 ± 1.67Bab	14.37 ± 1.08Bab	11.17 ± 1.27BCab	10.57 ± 1.45BCab	8.37 ± 0.93Cab	8.83 ± 0.89Cb	9.64 ± 0.98Cab
	60	24.62 ± 1.84Aa	18.38 ± 1.09Ba	17.98 ± 2.37BCa	14.65 ± 2.01BCDa	13.29 ± 1.68BCDa	10.12 ± 1.27Da	12.34 ± 1.38CDa	12.78 ± 1.37BCDa
10	0	23.93 ± 2.31Ab	17.96 ± 2.83Ba	17.29 ± 1.73BCa	14.82 ± 1.03BCb	13.23 ± 2.02BCa	11.83 ± 1.36BCb	11.20 ± 1.20Ca	11.11 ± 1.05Ca
	15	25.47 ± 1.87Ab	19.24 ± 1.14Ba	18.74 ± 1.46Ba	15.03 ± 1.27BCab	14.62 ± 1.78BCa	12.67 ± 0.98Cab	12.34 ± 1.44Ca	11.95 ± 1.18Ca
	30	26.99 ± 1.37Aab	21.65 ± 1.57Ba	19.26 ± 1.39BCa	15.98 ± 1.28CDab	15.24 ± 1.62CDa	13.46 ± 1.02Dab	13.18 ± 1.30Da	12.76 ± 1.33Da
	45	28.63 ±1.59Aab	22.97 ± 1.63Ba	20.98 ± 1.58BCa	16.74 ± 1.33CDab	16.32 ± 1.58CDa	14.79 ± 1.26Dab	14.67 ± 1.28Da	13.39 ± 1.49Da
	60	30.19 ± 1.62Aa	23.88 ± 1.77Ba	21.75 ± 2.01BCa	18.66 ± 1.49BCDa	17.95 ± 1.66CDa	16.03 ± 1.37CDa	15.29 ± 1.67Da	14.28 ± 1.52Da
15	0	30.82 ± 3.02Aa	23.93 ± 2.86AB	21.56 ± 3.19BCa	20.39 ± 2.63BCa	18.30 ± 2.53BCa	17.94 ± 1.94BCa	15.34 ± 1.50BCa	14.38 ± 1.44Ca
	15	31.24 ± 2.47Aa	25.13 ± 1.73Ba	22.91 ± 1.01BCa	21.57 ± 0.75BCDa	19.42 ± 1.10CDEa	19.02 ± 1.88CDEa	16.72 ± 0.47DEa	15.73 ± 1.32Ea
	30	33.19 ± 2.34Aa	26.79 ± 1.68Ba	24.04 ± 1.27BCa	22.82 ± 1.03BCDa	20.79 ± 1.78CDEa	19.87 ± 1.21CDEa	17.92 ± 0.87DEa	16.92 ± 1.29Ea
	45	34.68 ± 2.18Aa	27.36 ± 1.29Ba	25.69 ± 1.18BCa	24.29 ± 1.28BCa	21.94 ± 1.62CDa	20.75 ± 1.34CDa	18.37 ± 0.92Da	17.02 ± 1.45Da
	60	35.72 ± 2.57Aa	29.61 ± 1.11Ba	26.83 ± 1.37BCa	25.63 ± 1.39BCa	23.48 ± 1.56CDa	21.99 ± 1.55CDa	19.28 ± 1.25Da	18.54 ± 1.09Da
20	0	37.96 ± 3.60Aa	32.49 ± 3.25ABa	30.84 ± 2.87ABa	27.93 ± 2.62BCa	24.50 ± 2.43BCa	23.26 ± 2.30BCa	21.34 ± 1.95Ca	20.83 ± 2.01Ca
	15	38.88 ± 1.67Aa	33.68 ± 1a.09Aa	31.79 ± 1.78BCa	28.62 ± 1.73BCDa	25.50 ± 2.02CDa	24.69 ± 2.17CDa	22.68 ± 1.78Da	21.87 ± 1.76Da
	30	39.29 ± 2.04Aa	35.13 ± 1.23ABa	32.55 ± 1.69BCa	29.17 ± 1.38CDa	26.79 ± 1.38DEa	25.61 ± 1.68DEa	23.47 ± 1.64Ea	23.04 ± 1.58Ea
	45	40.76 ± 1.78Aa	36.62 ± 1.28ABa	33.49 ± 1.34BCa	30.56 ± 1.22CDa	27.15 ± 1.47DEa	27.01 ± 1.57DEa	24.39 ± 1.55Ea	24.65 ± 1.21Ea
	60	41.54 ± 1.46Aa	37.72 ± 1.36ABa	34.27 ± 1.22BCa	31.47 ± 1.35CDa	28.86 ± 1.58DEa	28.34 ± 1.34DEa	25.15 ± 1.28Ea	25.81 ± 1.33Ea

Values with the same letter are not statistically different, according to Duncan’s multiple range test at *p* < 0.05. a, b, c: mean values with the same letter in the same column are not significantly different. (*p* > 0.05) (*n* = 8). A, B, C, D: mean values with the same letter in the same row are not significantly different. (*p* > 0.05) (*n* = 8).

**Table 3 nanomaterials-10-01797-t003:** The strawberry preservation indexes of typical films in terms of weight loss ratio, firmness, content of total soluble solids and titration acid. (*p* > 0.05) (*n* = 6).

Strawberry Properties	Storage Time (Days)	Unpacked	PE	PVA/CMC	PVA/CMC/ ZnO NPs	PVA/CMC/*x*GnP	PVA/CMC/ ZnO NPs/*x*GnP
Weight loss ratio (%)	2	3.34 ± 0.11Ac	3.01 ± 0.53Ac	2.98 ± 0.47Ac	2.57 ± 0.77Ab	2.87 ± 1.04Ac	2.46 ± 0.27Ab
	4	12.92 ± 1.23Ab	12.58 ± 0.98Ab	11.79 ± 1.22ABb	8.54 ± 1.38ABb	10.86 ± 1.47ABb	7.51 ± 1.55Bb
	6	37.67 ± 1.54Aa	35.62 ± 1.68ABa	29.99 ± 2.49BCa	23.67 ± 1.79Ca	27.63 ± 1.82CDa	18.66 ± 1.99Da
Firmness(Normalized)	0	0.91964 ± 0.89706Aa	0.98214 ± 0.67647Aa	0.9375 ± 0.98529Aa	0.92857 ± 1Aa	0.94643 ± 0.10294Aa	1 ± 0.95588Aa
	2	0.6875 ± 0.39706Aa	0.79464 ± 0.83824Aa	0.73214 ± 0.52941Aa	0.77679 ± 0.55882Aa	0.75893 ± 0.67647Aa	0.80357 ± 0.70588Aa
	4	0.36607 ± 0.47059Aa	0.45536 ± 0.89706Aa	0.47321 ± 0.14706Aa	0.51786 ± 0.32353Aa	0.5 ± 0.60294Aa	0.55357 ± 0.97059Aa
	6	0 ± 0Aa	0.125 ± 0.02941ABa	0.1875 ± 0.11765ABa	0.28571 ± 0.23529ABa	0.24107 ± 0.22059ABa	0.41071 ± 0.17647Ba
Content of Total Soluble Solids (%)	0	11 ± 0.79Aa	11 ± 1.02Aa	12 ± 1.23Aa	11 ± 1.24Aa	11 ± 1.28Aa	12 ± 1.25Aa
	2	9 ± 0.23Aab	9.5 ± 0.67Aab	10 ± 0.92Aab	10 ± 0.92Aab	10.5 ± 1.05Ab	11 ± 1.38Aab
	4	6.5 ± 0.27Abc	7 ± 0.48ABbc	7.5 ± 0.38ABbc	8 ± 0.67Bbc	7 ± 0.89Bc	9.5 ± 0.79Bab
	6	4 ± 0.87Ac	5 ± 0.52ABc	6 ± 0.45ABc	6.5 ± 0.58ABCc	6.5 ± 0.57BCc	8 ± 0.35Cb
Titratable Acidity (%)	0	0.84 ± 0.21Aa	0.84 ± 0.22Aa	0.82 ± 0.14Aa	0.85 ± 0.17Aa	0.82 ± 0.01Aa	0.84 ± 0.12Aa
	2	0.75 ± 0.18Aa	0.76 ± 0.17Aa	0.75 ± 0.12Aa	0.79 ± 0.12Aa	0.75 ± 0.13Aa	0.80 ± 0.14Aa
	4	0.55 ± 0.11Aa	0.58 ± 0.12Aa	0.56 ± 0.05Aab	0.60 ± 0.07Aa	0.57 ± 0.06Aab	0.76 ± 0.08Aa
	6	0.31 ± 0.08Aa	0.33 ± 0.10Aa	0.33 ± 0.03Ab	0.42 ± 0.01Aa	0.35 ± 0.04Ab	0.53 ± 0.06Aa

Values with the same letter are not statistically different, according to Duncan’s multiple range test at *p* < 0.05. a, b, c: mean values with the same letter in the same column are not significantly different. (*p* > 0.05) (*n* = 6). A, B, C, D: mean values with the same letter in the same row are not significantly different. (*p* > 0.05) (*n* = 6).

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
