# Peer review of "Effects of Ultrasonication Time on the Properties of Polyvinyl Alcohol/Sodium Carboxymethyl Cellulose/Nano-ZnO/Multilayer Graphene Nanoplatelet Composite Films"

_nanomaterials, 2020, doi:10.3390/nano10091797_

Round 1
Reviewer 1 Report
The manuscript presents the fabrication and characterization of composite films with the aim to assess the role of ultrasonication during the preparation process. The study could be of interest, but, as a general concern, suffers from a number of drawbacks that can limit the potential outcome.
First of all, the readability is very poor, mostly due an exaggerated number of experimental conditions to be tested. This issue does not support a clear understanding of the collected findings. Related to the previous comment, the rationale of the study is unclear. Four different materials, with different ratios, were considered to fabricate the investigated composites. Was it strictly necessary to consider all these conditions to deal with a composite?
The Authors are strongly invited to shorten the manuscript in order to strengthen the final message.
In addition, the Results section should be modified in “Results and Discussion” and the Discussion section renamed “Conclusions”.
Finally, English language should be revised.
Specifically.
Materials and Methods
The degree of hydrolysis of PVA should be reported.
Par. 2.2 and 2.3 reported the preparation of polymeric and composite films, but, among the subsequent characterization tests, the possible role of the ultrasonication process on the degradation of PVA (e.g., molecular weight modification) and the water stability were not evaluated. The Authors should introduce ad hoc tests to address those potential issues.
It is not clear the need of a sedimentation experiment (par. 2.4), what is the expected added value of this test?
Regarding the density determination (par. 2.9), nothing is stated about the measurement accuracy of the size of the specimens to be tested, as opposed to the thickness.
In par. 2.11 it is reported “Mechanical properties were determined for bubble- and notch-free samples…”: what did the Authors mean? Moreover, did the Authors follow a standard?
Regarding the biodegradability assessment, it was stated “The results were reported as mean values for three samples of each group”, but nothing was reported about the experimental method used to collect data.
Results
The resolution of Fig. 1 is too low to be clearly observed.
The title of par. 3.2 is incorrect since there is no particle size distribution analysis.
In par. 3.3 the following sentence is reported “This finding also corroborated the results of tensile testing and morphology observations [34]”. This line is isolated from the text, it seems no more than a quote, and does not contribute to understand how data from FTIR analysis correlate to mechanical and morphological characteristics. Therefore, it can be removed or the topic needs to be examined in depth.
Moreover, Errata: Fig. 4; Corrige: Fig. 3 (line 308).
Young modulus of composite specimens was not reported, why? Please, add and discuss the requested data (par. 3.8)
In par. 3.12 a few points need to be clarified. Pictures of PVA, PVA/CMC, and PVA/CMC/ZnO NPs (day 0) show packed strawberries, while this occurrence was not verified for the other cases: why? Therefore, it seems that the visual inspection (other pictures) was carried out unpacking strawberries and then packing them again until the next time step: is it correct?
Most importantly: due to safety issues, did the Authors verify the absence of any nanomaterial transfer from films to strawberries?
Firmness values should be normalized in order to be coherently compared.
According to the text strawberries were warped, maybe they were wrapped.
Author Response
Comments and Suggestions for Authors 1:
The manuscript presents the fabrication and characterization of composite films with the aim to assess the role of ultrasonication during the preparation process. The study could be of interest, but, as a general concern, suffers from a number of drawbacks that can limit the potential outcome.
- First of all, the readability is very poor, mostly due an exaggerated number of experimental conditions to be tested. This issue does not support a clear understanding of the collected findings. Related to the previous comment, the rationale of the study is unclear. Four different materials, with different ratios, were considered to fabricate the investigated composites. Was it strictly necessary to consider all these conditions to deal with a composite?
Re: Thank you very much for your valuable comments. I have omitted some relatively insignificant test performance, such as stability, density and the light transmittance. The above conditions do not have to be strictly considered when evaluating the composite.
At present, most food packaging materials mainly consist of nonbiodegradable materials, which will cause irreversible environmental contaminations both in the long and short terms and are harmful to public health (M.F. Abou Taleb, H.L. Abd El-Mohdy, H.A. Abd El-Rehim, Radiation preparation of PVA/CMC copolymers and their application in removal of dyes, J Hazard Mater 168(1) (2009) 68-75.). The target of our work is to design a novel film for food packaging that have several advantages compared the commercially available films. Most of the commercial films have non-biodegradable and low mechanical properties and easily infected with microbes which, in turn, contaminate the preserved foods. In our work, the designed films contain PVA and CMC possessed good biodegradability, ZnO NPs and xGnP possessed low water vapor transmission rate and excellent antimicrobial properties and high mechanical properties. I chose these four different materials because there is no research on them at present. We explored the effects of different additions of zinc oxide and graphene on the properties of composite films, as well as the effects of different ultrasonic processing times on them. The optimal solution was obtained by this experiment, which was also verified by the experiment of strawberry preservation.
About the reasons for choosing these four different raw materials:
Yang et al. (J. Yang, Y. Zheng, L. Sheng, H. Chen, L. Zhao, W. Yu, K.-Q. Zhao, P. Hu, Water Induced Shape Memory and Healing Effects by Introducing Carboxymethyl Cellulose Sodium into Poly(vinyl alcohol), Industrial & Engineering Chemistry Research 57(44) (2018) 15046-15053.) found that polyvinyl alcohol (PVA) has been used for the preparation of packaging materials which have biodegradable property. It is often with carboxymethyl cellulose (CMC) to prepare composites (M.M. Abutalib, Effect of zinc oxide nanorods on the structural, thermal, dielectric and electrical properties of polyvinyl alcohol/carboxymethyle cellulose composites, Physica B: Condensed Matter 557 (2019) 108-116.). By doing this, the water resistance of PVA film can be improved. Therefore, we choose PVA and CMC (with strong hydrophilic ability) as the base material to develop degradable polymer films for food packaging. There are many research studies in the literature employed polymers like carboxymethyl cellulose (CMC) and polyvinyl alcohol (PVA) as film forming material (H.F. Youssef, M.E. El-Naggar, F.K. Fouda, A.M. Youssef, Antimicrobial packaging film based on biodegradable CMC/PVA-zeolite doped with noble metal cations, Food Packaging and Shelf Life 22 (2019) 100378.). However, pure PVA/CMC composite films feature low water resistance and have no resistance to pathogens, which has inspired many works on antimicrobial activity enhancement via the incorporation of nanoparticulate fillers. Currently, ZnO NPs is one of the five zinc compounds that are listed as a generally recognized as safe (GRAS) material by the U.S. Food and Drug Administration (P.J.P. Espitia, N.d.F.F. Soares, J.S.d.R. Coimbra, N.J. de Andrade, R.S. Cruz, E.A.A. Medeiros, Zinc Oxide Nanoparticles: Synthesis, Antimicrobial Activity and Food Packaging Applications, Food and Bioprocess Technology 5(5) (2012) 1447-1464.). What is more, ZnO NPs has high catalytic activity against chemical and biological species and strong antimicrobial properties against a wide range of pathogens (K. Lefatshe, C.M. Muiva, L.P. Kebaabetswe, Extraction of nanocellulose and in-situ casting of ZnO/cellulose nanocomposite with enhanced photocatalytic and antibacterial activity, Carbohydr Polym 164 (2017) 301-308.). Zhong et al. (R. Zhong, Q. Zhong, M. Huo, B. Yang, H. Li, Preparation of biocompatible nano-ZnO/chitosan microspheres with multi-functions of antibacterial, UV-shielding and dye photodegradation, Int J Biol Macromol 146 (2020) 939-945.) attempted to incorporate ZnO NPs into a polymer to fabricate multi-functional ZnO NPs/CS composite with enhanced biosafety. In addition, its thermal and antimicrobial properties are improved, whereas mechanical properties experience deterioration due to the NP surface effect and interfacial action (T.-j. Zhou, Y.-y. Hu, R.-y. Chen, X. Zheng, X. Chen, Z. Chen, J.-q. Zhong, Preparation and characterization of bipolar membranes modified by photocatalyst nano-ZnO and nano-CeO2, Applied Surface Science 258(8) (2012) 4023-4027.). Hence, in order to enhance the mechanical properties of the film, we took the method of combining ZnO NPs with graphene to obtain superior composites, as the addition of appropriate amounts of graphene to polymers can significantly improve their mechanical properties through the effects of van der Waals forces.
- The Authors are strongly invited to shorten the manuscript in order to strengthen the final message.
Re: First of all, thank you very much for your valuable comments. In response to your comments, I have omitted some relatively insignificant test performance, such as stability, density and the light transmittance. At the same time, I put the performance data of light transmittance in the appendix for reference. Thanks.
- In addition, the Results section should be modified in “Results and Discussion” and the Discussion section renamed “Conclusions”.
Re: Corrected, thanks.
- Finally, English language should be revised.
Re: Some of the errors in this article have been modified. Thanks.
Specifically.
Materials and Methods
- The degree of hydrolysis of PVA should be reported.
Re: Corrected, PVA (1700 degree of polymerization and 99% hydrolyzation). Thanks.
- 2.2 and 2.3 reported the preparation of polymeric and composite films, but, among the subsequent characterization tests, the possible role of the ultrasonication process on the degradation of PVA (e.g., molecular weight modification) and the water stability were not evaluated. The Authors should introduce ad hoc tests to address those potential issues.
Re: Thanks for your kind and suggestive comments. Firstly, in terms of molecular weight modification, I'm sorry that our school is unable to carry out the molecular weight test. So I looked up the relevant literature. Abral. et al (H. Abral, A. Atmajaya, M. Mahardika, F. Hafizulhaq, Kadriadi, D. Handayani, S.M. Sapuan, R.A. Ilyas, Effect of ultrasonication duration of polyvinyl alcohol (PVA) gel on characterizations of PVA film, Journal of Materials Research and Technology 9(2) (2020) 2477-2486.) found that the viscosity of PVA gel after ultrasonication is lower than that before ultrasonication. The decrease in this viscosity is because due to depolymerisation and decreased the molecular weight of PVA chains. The shorter PVA chains are free more mobile that can restructure their chains themselves during the drying process, so resulting in more compact and homogenous polymer structure after ultrasonication. The depolymerized PVA chains due to ultrasonication also have a higher amount of free hydroxyl groups]. A similar finding was also reported by Mohod. et al (A.V. Mohod, P.R. Gogate, Ultrasonic degradation of polymers: effect of operating parameters and intensification using additives for carboxymethyl cellulose (CMC) and polyvinyl alcohol (PVA), Ultrason Sonochem 18(3) (2011) 727-34.) that with the use of sonication, viscosity decreased significantly for the polymers.
Secondly, the water stability of the composite film is mentioned. In the previous experiment of this part, due to the good water solubility of the composite film, it can not be completely removed and weighed. So we didn't adopt it. Now, In order to meet the requirements of reviewers, we supplemented this part of the experiment. The details are as follows:
The films were cut into 3 cm×3 cm pieces for the determination of solubility and swelling degree. The pieces were dried at 105 ℃ to constant weight to obtain the initial dry mass (M1). Then, they were placed in 100 mL beakers with 50 mL distilled water covered with plastic wraps and stored at 25 ℃ for 24 h.
Figure 6 shows the dissolution and swelling condition of PVA/CMC, PVA/CMC/ZnO NPs, PVA/CMC/ZnO NPs/xGnP (ZnO NPs:xGnP = 9:1, 7:3, 5:5, 3:7, 1:9), PVA/CMC/xGnP film with 30 min ultrasonication and PVA/CMC/ZnO NPs/xGnP (ZnO NPs:xGnP = 7:3) composite films under different ultrasonication times. During the swelling experiment, all films were wrinkled when they came into contact with water. And the most severe degree of shrinkage was PVA/CMC film. PVA is well known that it has high degree of swelling in aqueous solvents (R. Agarwal, M.S. Alam, B. Gupta, Polyvinyl alcohol-polyethylene oxide-carboxymethyl cellulose membranes for drug delivery, Journal of Applied Polymer Science 129(6) (2013) 3728-3736.). Similarly, CMC also has good hydrophilicity. In addition, we can observe that there are holes in the PVA/CMC/xGnP film, possibly as a result of the formation of extra pores on the polymer surface due to the ultrasonication (K. Spilarewicz-Stanek, A. Kisielewska, J. Ginter, K. Bałuszyńska, I. Piwoński, Elucidation of the function of oxygen moieties on graphene oxide and reduced graphene oxide in the nucleation and growth of silver nanoparticles, RSC Advances 6(65) (2016) 60056-60067.). It could also be because that xGnP is prone to agglomeration under the action of van der Waals forces and π-π stacking interactions, which leads to uneven dispersion of the solution and the appearance of pores.
Figure 6. The dissolution and swelling condition of PVA/CMC, PVA/CMC/ZnO NPs, PVA/CMC/ZnO NPs/xGnP (ZnO NPs:xGnP = 9:1, 7:3, 5:5, 3:7, 1:9), PVA/CMC/xGnP film with 30 min ultrasonication and PVA/CMC/ZnO NPs/xGnP (ZnO NPs:xGnP = 7:3) composite films under different ultrasonication times.
- It is not clear the need of a sedimentation experiment (par. 2.4), what is the expected added value of this test?
Re: I think there are two reasons why the sedimentation experiment is not clear. Firstly, graphene is black, so the dispersion of the solution is not obvious. Moreover, the experiment time is too short, resulting in the incomplete dispersion. Thanks.
- Regarding the density determination (par. 2.9), nothing is stated about the measurement accuracy of the size of the specimens to be tested, as opposed to the thickness.
Re: Added. Thanks.
- In par. 2.11 it is reported “Mechanical properties were determined for bubble- and notch-free samples…”: what did the Authors mean? Moreover, did the Authors follow a standard?
Re: When preparing and cutting the composite films, bubbles and notches sometimes appear due to improper operation. Their presence has an effect on the outcome, making the break point artificially appear early. So we're going to avoid it happening. We have followed ASTM. (2010) standard (Standard test methods for tensile properties of thin plastic sheeting. D882–10. In Annual book of ASTM. Philadelphia, PA: American Society for testing and Materials. 2010.). Thanks.
- Regarding the biodegradability assessment, it was stated “The results were reported as mean values for three samples of each group”, but nothing was reported about the experimental method used to collect data.
Re: Added. Thanks. The details are as follows: Biodegradability was determined by measuring the weight loss of the membranes buried under soil (R. Zhang, Y. Wang, D. Ma, S. Ahmed, W. Qin, Y. Liu, Effects of ultrasonication duration and graphene oxide and nano-zinc oxide contents on the properties of polyvinyl alcohol nanocomposites, Ultrason Sonochem 59 (2019) 104731.). For the biodegradability analysis, the membranes were cut into 6 cm×1 cm piece (another 3 cm × 3cm samples were taken for photographing), weighed, tied with one corner with a thread and buried at about 15 cm below the surface of the soil. The buried membranes were removed from the soil every two weeks, washed by deionized water, dried at 60 ℃ until the weight of the films didn’t change. The weight loss was then calculated using the following equation:
where, Wi=initial weight of the specimen and Wd=dry weight of the specimen after degradation in soil.
Results
- The resolution of Fig. 1 is too low to be clearly observed.
Re: Thanks for your kind and suggestive comments. I think there are three reasons why the sedimentation experiment is not clear. Firstly, graphene is black, so the dispersion of the solution is not obvious. Moreover, the experiment time is too short, resulting in the incomplete dispersion. Another possibility is that the image has been compressed, resulting in poor resolution. As the sample bottle of settlement experiment was retained before, we reshot this part. However, its dispersion is not clear enough, we choose to put it in the appendix for reference.
- The title of par. 3.2 is incorrect since there is no particle size distribution analysis.
Re: Thanks for your kind and suggestive comments. Corrected, thanks.
- In par. 3.3 the following sentence is reported “This finding also corroborated the results of tensile testing and morphology observations [34]”. This line is isolated from the text, it seems no more than a quote, and does not contribute to understand how data from FTIR analysis correlate to mechanical and morphological characteristics. Therefore, it can be removed or the topic needs to be examined in depth.
Re: Thanks for your kind and suggestive comments. Removed, thanks.
- Moreover, Errata: Fig. 4; Corrige: Fig. 3 (line 308).
Re: Thanks for your kind and suggestive comments. Corrected, thanks.
- Young modulus of composite specimens was not reported, why? Please, add and discuss the requested data (par. 3.8)
Re: Thanks for your kind and suggestive comments. In response to your comments, we have supplemented. Thanks. The details are as follows:
The tensile strength of a material reflects its resistance to the breaking process when a constant load is applied on the material. Another significant mechanical property is Young’s modulus, also called as elastic modulus, which is a property of linear elastic solid materials. This property describes the relationship between stress and strain of a material (A. Goswami, A.K. Bajpai, J. Bajpai, B.K. Sinha, Designing vanadium pentoxide-carboxymethyl cellulose/polyvinyl alcohol-based bionanocomposite films and study of their structure, topography, mechanical, electrical and optical behavior, Polymer Bulletin 75(2) (2017) 781-807.). Figure 8c shows that the addition of ZnO NPs and xGnP significantly affected the Young's modulus. In combination with Figure 8b, the variation trend of tensile strength of the composite film is basically the same as the stress (p < 0.05). Pure PVA/CMC films featured a maximum tensile strength of 45.50 ± 3.18 MPa and a maximum Young's modulus of 33.95 ± 2.93 MPa, which was hardly affected by the incorporation of only ZnO NPs or graphene. This finding was ascribed to the poor compatibility between ZnO NPs or xGnP and PVC/CMC, which resulted in weak interfacial adhesion between the two phases, and hence, in the easy initiation and propagation of cracking at the corresponding interface (Y.G. Zhou, X.D. Zhao, B.B. Dong, C.T. Liu, Improvement of the dispersity of micro‐nano particles for PP/PVC composites using gas‐assisted dispersion in a controlled foaming process, Polymer Engineering & Science 60(3) (2019) 524-534.). However, this impact was small. At a ZnO NPs:xGnP ratio of 9:1, the Young's modulus of the composite film sharply decreased to 2.61 ± 0.62 MPa, which was 92.14% lower than that of the pure PVA/CMC film (33.21 ± 2.90 MPa), i.e., the addition of xGnP reduced the plasticity of the material. This behavior was in line with the results of Spilarewicz-Stanek. et al. (S. Ranjan, B. Mukherjee, A. Islam, K.K. Pandey, R. Gupta, A.K. Keshri, Microstructure, mechanical and high temperature tribological behaviour of graphene nanoplatelets reinforced plasma sprayed titanium nitride coating, Journal of the European Ceramic Society 40(3) (2020) 660-671.) and was ascribed to the antagonism between crack bridging, crack arrest, and strong interface formation.
Figure 8. Strain (a), Stress (b) and Young's modulus (c) of pure PVA/CMC film, PVA/CMC/ZnO NPs/xGnP composite films under different ultrasonication times. (*: p < 0.05, n = 8).
- In par. 3.12 a few points need to be clarified. Pictures of PVA, PVA/CMC, and PVA/CMC/ZnO NPs (day 0) show packed strawberries, while this occurrence was not verified for the other cases: why? Therefore, it seems that the visual inspection (other pictures) was carried out unpacking strawberries and then packing them again until the next time step: is it correct?
Re: Thanks for your kind and suggestive comments. In the preparation of composite membranes, as the addition of appropriate amounts of graphene to polymers can significantly improve their mechanical properties through the effects of van der Waals forces. Parameswaranpillai et al (J. Parameswaranpillai, M.R. Sanjay, S.A. Varghese, S. Siengchin, S. Jose, N. Salim, N. Hameed, A. Magueresse, Toughened PS/LDPE/SEBS/xGnP ternary composites: morphology, mechanical and viscoelastic properties, International Journal of Lightweight Materials and Manufacture 2(1) (2019) 64-71.) have shown that the incorporation of exfoliated graphene nanoplatelets (xGnP) in ternary PP/PS/SEBS blends significantly improved the tensile modulus and elongation at break of the ternary. But it is precisely because of the addition of graphene, so that the color of the composite film is black. And in the strawberry preservation test, we cannot directly see the changes in strawberries. So we carried out unpacking strawberries.
- Most importantly: due to safety issues, did the Authors verify the absence of any nanomaterial transfer from films to strawberries?
Re: Thanks for your kind and suggestive comments. We also took this issue into consideration when designing the experiment, but unfortunately our experimental conditions were unable to quantitatively analyze the migration of nanomaterials. We consulted a series of related documents in the early stage of the experiment and found that the migration number of nanomaterials was within the scope of safety standards. For example, Heydari-Majd et al. studied the content of Zn2+ ions in fish fillets wrapped in PLA/ZnO composite membranes, and found that the migration amounts of Zn2+ ions from the nanocomposite membrane to fillet samples up to 1.551±0.160 mg/100 g sample, which was still far below the migration limit of 40 mg/day on zinc daily consumption as defined by the National Institute of Health for food contact materials (https://ods.od.nih.gov/factsheets/Zinc-HealthProfessional/) (M. Heydari-Majd, B. Ghanbarzadeh, M. Shahidi-Noghabi, M. Ali Najafi, M. Hosseini, A new active nanocomposite film based on PLA/ZnO nanoparticle/essential oils for the preservation of refrigerated Otolithes ruber fillets, Food Packaging and Shelf Life 19 (2019) 94-103.). Panea et al. analyzed the migration of ZnO and Ag particles in the aqueous food simulant were analyzed by inductively coupled plasma mass spectrometry (ICP-MS), and found that the migration of nanoparticles in the simulant was very low, and the migration of Zn2+ in the control package was below the detection limit (<0.005 mg /kg), and in the packaging of the added nanoparticles, only (2.44 ± 0.37) mg /kg of Zn2+ concentration was detected, which is well below the limit established by COMMISSION REGULATION (EU) No 10/2011 (25 mg/kg food or food simulant) (B. Panea, G. Ripoll, J. González, A. Fernández-Cuello, P. Alberti, Effect of nanocomposite packaging containing different proportions of ZnO and Ag on chicken breast meat quality, Journal of Food Engineering 123 (2014) 104-112.). Thanks.
- Firmness values should be normalized in order to be coherently compared.
Re: Thanks for your kind and suggestive comments. Corrected, thanks.
- According to the text strawberries were warped, maybe they were wrapped.
Re: As you can see, strawberries are wrapped. The growth and reproduction of microorganisms on the strawberries bringing out the rapid decay of fresh fruits. Therefore, a small amount of juice is seeped out, causing the composite film to wrinkle slightly. At the same time, the crumpled membrane reverses to the strawberry, causing the strawberry to twist.
Reviewer 2 Report
The article "Effects of ultrasonication time on the properties of polyvinyl alcohol /sodium carboxymethyl cellulose/nano-ZnO/multilayer graphene nanoplatelet composite films" describes the synthesis and characterization of antibacterial food packaging material based on CMC, PVA, ZnO and graphene.
The abbreviation must be given only once. Row 37, 38 and 43 presents again PVA meaning.
The notations must be consistent (e.g. mL, while at row 177 are both mL and ml).
Meaning of some sentences eludes me (row 187-188).
Is very hard to see any differences due to ultrasonication time in fig 1.
For fig.3 a presentation side by side in two groups I think is more appropriate. The figure is too long and resizing it will impact the legibility.
The DSC analysis has some flawed interpretations. First of all I fail to see the mentioned exothermic peaks in the presented DSC curves. Maybe with a better resolution picture, with numeric data for heat flow authors could sustain their affirmation. As I see the picture (DSC of PVA/CMC) the “hill” between two endothermic effects is just the baseline from which endothermic effects are going down.
I do see on PVA/CMC film a small exothermic peak at about 280-290oC, which can be assigned to crystalline transition. Other exothermic processes under N2 atmosphere are highly unlikely.
The authors have put in table values for Tc and Xc with no explanation of what they mean.
A glass transition temperature between 90-95oC for the bare polymer mix is in line with literature values (see DOI: 10.1016/j.physb.2011.07.050). For the rest of composites, the nanoparticles can induce an increase of the Tg.
The melting temperatures are identified wrongly. PVA has a melting point around 217oC. For similar blends PVA/CMC literature reports a m.p. of ~ 218oC, therefore the melting process is what authors have noted by Tc at 226-227oC. At 294-313oC the literature reports the decomposition of the polymer mix (where TG/DSC data are available as from DSC data only the authors have no way to see the mass loss). The melting process can be regarded largely as unaffected by ZnO or graphene (by looking at Tc data provided by authors). By contrast the decomposition temperature is affected by strong interactions with ZnO nanoparticles but not by the inert graphene.
At row 372 is mentioned figure 5 (DSC) and 26.48o (probably from XRD).
At row 394 the values can’t be right 0.12±0.13 g/cm3
At row 482 S. et al. [44] proper author name must be given.
Author Response
Comments and Suggestions for Authors 2:
The article "Effects of ultrasonication time on the properties of polyvinyl alcohol /sodium carboxymethyl cellulose/nano-ZnO/multilayer graphene nanoplatelet composite films" describes the synthesis and characterization of antibacterial food packaging material based on CMC, PVA, ZnO and graphene.
- The abbreviation must be given only once. Row 37, 38 and 43 presents again PVA meaning.
Re: Thanks for your kind and suggestive comments. Removed, thanks.
- The notations must be consistent (e.g. mL, while at row 177 are both mL and ml).
Re: Thanks for your kind and suggestive comments. Corrected, thanks.
- Meaning of some sentences eludes me (row 187-188).
Re: Since strawberries can't be bought locally in July, we ordered them online. A batch of fresh strawberries (purchased from Dandong, Liaoning) were transported by cold chain. Thanks.
- Is very hard to see any differences due to ultrasonication time in fig 1.
Re: I think there are two reasons why the sedimentation experiment is not clear. Firstly, graphene is black, so the dispersion of the solution is not obvious. Moreover, the experiment time is too short, resulting in the incomplete dispersion. Thanks.
- For fig.3 a presentation side by side in two groups I think is more appropriate. The figure is too long and resizing it will impact the legibility.
Re: Corrected, thanks.
- The DSC analysis has some flawed interpretations. First of all I fail to see the mentioned exothermic peaks in the presented DSC curves. Maybe with a better resolution picture, with numeric data for heat flow authors could sustain their affirmation. As I see the picture (DSC of PVA/CMC) the “hill” between two endothermic effects is just the baseline from which endothermic effects are going down.
Re: We've eliminated exothermic peaks that don't exist. Thanks.
- I do see on PVA/CMC film a small exothermic peak at about 280-290oC, which can be assigned to crystalline transition. Other exothermic processes under N2 atmosphere are highly unlikely.
Re: After checking again, we added the small exothermic peak at 289.65 °C on PVA/CMC film. Thanks.
- The authors have put in table values for Tc and Xc with no explanation of what they mean.
Re: Added. Tg: transition temperature of the plate glass, Tc: critical temperature, Tm: melting temperature, Xc: capacitive reactance. Thanks.
- A glass transition temperature between 90-95oC for the bare polymer mix is in line with literature values (see DOI: 1016/j.physb.2011.07.050). For the rest of composites, the nanoparticles can induce an increase of the Tg.
Re: I have read this article and quoted it in the article. Thanks.
- The melting temperatures are identified wrongly. PVA has a melting point around 217oC. For similar blends PVA/CMC literature reports a m.p. of ~ 218oC, therefore the melting process is what authors have noted by Tc at 226-227oC. At 294-313oC the literature reports the decomposition of the polymer mix (where TG/DSC data are available as from DSC data only the authors have no way to see the mass loss). The melting process can be regarded largely as unaffected by ZnO or graphene (by looking at Tc data provided by authors). By contrast the decomposition temperature is affected by strong interactions with ZnO nanoparticles but not by the inert graphene.
Re: Corrected, thanks.
- At row 372 is mentioned figure 5 (DSC) and 26.48o (probably from XRD).
Re: Corrected, thanks.
- At row 394 the values can’t be right 0.12±0.13 g/cm3
Re: Corrected, thanks.
- At row 482 S. et al. [44] proper author name must be given.
Re: Corrected, thanks.
Reviewer 3 Report
Ji et al. describes the preparation of a PVA/CMC/nanoZnO/xGNP composite films via sonication and investigated the effects of nanoZnO:xGNP mass ratio and sonication time on the performance of the films in terms of a wide variety of parameters. The long list of parameters broadens the scope of the paper, so much so that it obscures the focus of the work. The paper is heavily data-centric with minimal discussion - the discussion section at the end of the paper is too brief vis-a-vis the data content that has been presented. It also lacks a Conclusion section. While a lot of the data is undoubtedly beneficial for others pursuing the same kind of work, it is practically of little merit for others working with other materials and using different approaches. I think the authors need to seriously reorganize the data presented in this work so that it would be more focused yet benefits a wider readership.
Author Response
Comments and Suggestions for Authors 3:
Ji et al. describes the preparation of a PVA/CMC/nanoZnO/xGNP composite films via sonication and investigated the effects of nanoZnO:xGNP mass ratio and sonication time on the performance of the films in terms of a wide variety of parameters. The long list of parameters broadens the scope of the paper, so much so that it obscures the focus of the work. The paper is heavily data-centric with minimal discussion - the discussion section at the end of the paper is too brief vis-a-vis the data content that has been presented. It also lacks a Conclusion section. While a lot of the data is undoubtedly beneficial for others pursuing the same kind of work, it is practically of little merit for others working with other materials and using different approaches. I think the authors need to seriously reorganize the data presented in this work so that it would be more focused yet benefits a wider readership.
Re: I have omitted some relatively insignificant test performance, such as stability, density and the light transmittance. Thanks.
About the focus of the work:
The target of our work is to design a novel film for food packaging that have several advantages compared the commercially available films. Most of the commercial films have non-biodegradable and low mechanical properties and easily infected with microbes which, in turn, contaminate the preserved foods. In our work, the designed films contain PVA and CMC possessed good biodegradability, ZnO NPs and xGnP possessed low water vapor transmission rate and excellent antimicrobial properties and high mechanical properties. I chose these four different materials because there is no research on them at present. We explored the effects of different additions of zinc oxide and graphene on the properties of composite films, as well as the effects of different ultrasonic processing times on them. The optimal solution was obtained by this experiment, which was also verified by the experiment of strawberry preservation.
About the focus of the Conclusion section:On the basis of the reviewer 1's comments, I renamed the Discussion to Conclusions. At the same time, I supplemented the original Discussion and added some data to support the conclusions of this experiment. The details are as follows:
Conclusion: The target of our work is to design a novel film for food packaging. In our work, PVA/CMC/ZnO NPs/xGnP composite films were successfully fabricated by ultrasonication and solution casting, and the best overall performance (e.g., strong antibacterial activity against both Gram-positive and Gram-negative foodborne pathogenic bacteria. And the best preservation effect on strawberries: the strawberries warped in PVA/CMC/ZnO NPs/xGnP (7:3) films have the least weight loss, 19.01% less than the unpacked groups on the sixth day.) was observed at a ZnO NPs:xGnP mass ratio of 7:3. In addition, the PVA/CMC/ZnO NPs/xGnP composite film (ZnO NPs:xGnP mass ratio of 7:3) has low water vapor transmission rate. By studying the effects of different ultrasonic times on the properties of composite membranes, we found that ultrasonication improved the mechanical properties of composite film and in combination with traditional film preparation methods, at an ultrasonication time of 30 min, the pure PVA/CMC film exhibited a strain of 165.60 ± 8.88%, which was 31.57% higher than that observed for the non-ultrasonicated film. At the same time, ultrasonic treatment also significantly improves biodegradability. In a word, it was concluded to be well suited for the production of composite films with enhanced mechanical, antibacterial activity, biodegradability and barrier properties for commercial food packaging applications.
Round 2
Reviewer 1 Report
The Authors addressed most of the concerns. A few points should be further considered to definitely improve the manuscript.
Par. 2.1. The Authors removed PVA Mw and Mw/Mn, why? Please, add these data.
Par. 2.9 The Authors added the size of the specimens to be tested, but not the related measurement accuracy, why?
Par. 2.11 The ASTM standard should be added
Par. 3.7. The following sentence seems incorrect as the Authors aimed to describe the same trend of tensile strength and Young’s modulus “In combination with Figure 8b, the variation trend of tensile strength of the composite film is basically the same as the stress”.
Par. 3.12. The reply to the safety issues should be added to the discussion.
Author Response
Comments and Suggestions for Authors 1:
The Authors addressed most of the concerns. A few points should be further considered to definitely improve the manuscript.
- The Authors removed PVA Mw and Mw/Mn, why? Please, add these data.
Re: Thank you very much for your valuable comments. Added, thanks.
- 2.9 The Authors added the size of the specimens to be tested, but not the related measurement accuracy, why?
Re: Thank you very much for your valuable comments. Added, thanks.
- 2.11 The ASTM standard should be added
Re: Thank you very much for your valuable comments. Added, thanks.
- 3.7. The following sentence seems incorrect as the Authors aimed to describe the same trend of tensile strength and Young’s modulus “In combination with Figure 8b, the variation trend of tensile strength of the composite film is basically the same as the stress”.
Re: Thank you very much for your valuable comments. Corrected, thanks.
- 3.12. The reply to the safety issues should be added to the discussion.
Re: Thank you very much for your valuable comments. Added, thanks.
Reviewer 3 Report
The content is still lengthy and can still be further streamlined; for example the multiple subheadings on the various specific material properties can be categorized under broader headings like chemical property, biodegradability, optical property, etc. And as previously mentioned there needs to be some kind contribution to a wider readership. At present, all insights presented are very specifically applicable to this particular composite material. How can others, who will almost definitely work with other composite materials benefit from the information presented in this work? The authors have yet to address this crucial point.
Author Response
Comments and Suggestions for Authors 2:
The content is still lengthy and can still be further streamlined; for example the multiple subheadings on the various specific material properties can be categorized under broader headings like chemical property, biodegradability, optical property, etc. And as previously mentioned there needs to be some kind contribution to a wider readership. At present, all insights presented are very specifically applicable to this particular composite material. How can others, who will almost definitely work with other composite materials benefit from the information presented in this work? The authors have yet to address this crucial point.
Re: Thank you very much for your valuable comments. The answers to the relevant questions are as follows. Thanks.
- First of all, we have made appropriate abridgement for Mechanical properties, Antimicrobial activity. In addition, we've taken your advice, and we've included the multiple subheadings on the various specific material properties in broader headings. The details are as follows:
Chemical property - Swelling property
Physical property
Composite morphology
ATR-FTIR analysis
XRD analysis
DSC analysis
WVP
Mechanical properties
Surface color
Antimicrobial activity
Safety issues
Biodegradability
Preservation experiment of strawberries
Photographs of strawberries during the storage time
Strawberry properties
- Regarding the reference value of this research to others, I think there are mainly the following three aspects.
- Our study is of reference value in exploring the effects of ZnO NPs and xGnP fillers on the properties of composite films. At present, there are relatively few literatures on the effect of multi-nano fills on membrane properties. In our experiment, we set up several groups to study the effects of ZnO NPs and xGnP on composite films and the effects of their combined effects on composite films. Moreover, our experiments have gained some regularities. For mechanical properties, the addition of ZnO NPs or xGnP significantly increased the elongation at break and reduced the young's modulus, which is consistent with Rodríguez-Tobías. et al ( Rodríguez-Tobías, G. Morales, D. Grande, Improvement of mechanical properties and antibacterial activity of electrospun poly( d , l -lactide)-based mats by incorporation of ZnO- graft -poly( d , l -lactide) nanoparticles, Materials Chemistry and Physics 182 (2016) 324-331.) and Scaffaro. et al (R. Scaffaro, L. Botta, A. Maio, G. Gallo, PLA graphene nanoplatelets nanocomposites: Physical properties and release kinetics of an antimicrobial agent, Composites Part B: Engineering 109 (2017) 138-146.), respectively. The lowest elongation at break was observed at a ZnO NPs:xGnP ratio of 5:5, possibly because the fillers themselves and especially the filler-matrix interface are stressed and may lose structural integrity, which may be regarded as the formation of holes in the matrix to create initial fracture (K. Spilarewicz-Stanek, A. Kisielewska, J. Ginter, K. Bałuszyńska, I. Piwoński, Elucidation of the function of oxygen moieties on graphene oxide and reduced graphene oxide in the nucleation and growth of silver nanoparticles, RSC Advances 6(65) (2016) 60056-60067.). In terms of water vapor transmittance, WVP decreases after the introduction of ZnO NPs or xGnP alone. The result of adding ZnO NPs is consistent with that of Arfat. et al (Y.A. Arfat, S. Benjakul, T. Prodpran, P. Sumpavapol, P. Songtipya, Properties and antimicrobial activity of fish protein isolate/fish skin gelatin film containing basil leaf essential oil and zinc oxide nanoparticles, Food Hydrocolloids 41 (2014) 265-273.). And Clausi. et al used PVDF and GnPs to fabricate Hydrophobic polymeric nanocomposite coatings with high thermal conductivity (M. Clausi, S. Grasselli, A. Malchiodi, I.S. Bayer, Thermally conductive PVDF-graphene nanoplatelet (GnP) coatings, Applied Surface Science 529 (2020) 147070.). However, there are few studies on the simultaneous use of ZnO and xGnP to prepare composite films. Therefore, more similar studies are needed to further explore.
- In this experiment, we performed ultrasound treatment on the composite films. Ultrasonication is widely employed in the field of materials design and engineering, as its heating, mechanical, super-mixing, and cavitation effects facilitate the breakage of the original hydrogen bonds and hydrophobic bonds in the medium and thus promote the formation and exposure of more reaction centers ( Wang, N. Ma, Y. Yan, Z. Wang, Ultrasonic-assisted fabrication of high flux T-type zeolite membranes on alumina hollow fibers, Journal of Membrane Science 548 (2018) 676-684.). In turn, this accelerates chemical reactions and allows macromolecules to recombine and form a variety of nonpolar bonds. Moreover, ultrasonication further improves material solubility and makes films adopt more compact network structures (M. Asrofi, H. Abral, Y.K. Putra, S.M. Sapuan, H.-J. Kim, Effect of duration of sonication during gelatinization on properties of tapioca starch water hyacinth fiber biocomposite, International Journal of Biological Macromolecules 108 (2018) 167-176). We found that after 30 min of ultrasonication, the WVP of the PVA membrane increase, which is consistent with Zhang. et al (R. Zhang, Y. Wang, D. Ma, S. Ahmed, W. Qin, Y. Liu, Effects of ultrasonication duration and graphene oxide and nano-zinc oxide contents on the properties of polyvinyl alcohol nanocomposites, Ultrason Sonochem 59 (2019) 104731.). While the WVP of the composite nanofibrous membrane increased,but the increase is less than that of the PVA/CMC film. The change trend of WVP of composite films under 30 min of ultrasonication is contrary to the conclusion of Zhang. et al (R. Zhang, Y. Wang, D. Ma, S. Ahmed, W. Qin, Y. Liu, Effects of ultrasonication duration and graphene oxide and nano-zinc oxide contents on the properties of polyvinyl alcohol nanocomposites, Ultrason Sonochem 59 (2019) 104731.), possibly due to the different nanofillers selected. Therefore, it can be concluded that under the ultrasound treatment for 30 minutes, ZnO NPs and GO are more closely integrated than xGnP, and ZnO NPs and GO fibers are more uniformly dispersed in the matrix. At present, ultrasonic treatment membrane is still relatively few. The current research can provide a certain basis for the subsequent introduction of ultrasound in composite film.
- We applied the composite film to food preservation, combined the experiment with practice, and strived to maximize the application.
In our work, PVA/CMC/ZnO NPs/xGnP composite films were successfully fabricated, and the best preservation effect on strawberries (the strawberries warped in PVA/CMC/ZnO NPs/xGnP (7:3) films have the least weight loss, 19.01% less than the unpacked groups on the sixth day.) was observed at a ZnO NPs:xGnP mass ratio of 7:3. The film samples containing ZnO NPs displayed inhibition against both investigated bacteria. The antibacterial activity of ZnO NPs is related to the disruption of bacterial cell membranes by Zn2+ ions and oxidative stress owing to the production of reactive oxygen species (ROS) by ZnO NPs (S. Amjadi, M. Nazari, S.A. Alizadeh, H. Hamishehkar, Multifunctional betanin nanoliposomes-incorporated gelatin/chitosan nanofiber/ZnO nanoparticles nanocomposite film for fresh beef preservation, Meat Sci 167 (2020) 108161.). At present, there are few studies on the preservation of food by composite films containing xGnP, mainly focusing on the effects of xGnP on the properties of composite films. Ebrahimzadeh. et al (S. Ebrahimzadeh, B. Ghanbarzadeh, H. Hamishehkar, Physical properties of carboxymethyl cellulose based nano-biocomposites with Graphene nano-platelets, Int J Biol Macromol 84 (2016) 16-23.) found that at the low content of filler, GNPs probably dispersed well in the CMC matrix and blocked the water vapor transmission. However, additional amount of GNPs might agglomerate which in turn increased empty spaces and facilitated the water vapor transmission. This performance of xGnP is conducive to the food preservation, which is also reflected in our experiments.
At present, the composite film containing nano-fillers is used less in food preservation, especially the two nano-fillers. Al-Tayyar. et al (N.A. Al-Tayyar, A.M. Youssef, R.R. Al-Hindi, Antimicrobial packaging efficiency of ZnO-SiO2 nanocomposites infused into PVA/CS film for enhancing the shelf life of food products, Food Packaging and Shelf Life 25 (2020).) found that the prepared packaging material containing 5% ZnO-SiO2 nanocomposite prevented mold growth on the bread surface and exhibited the highest antibacterial activity against Gram-positive bacteria (Staphylococcus aureus) and Gram-negative bacteria (Escherichia coli) according to the colony forming unit technique. This experiment can provide a certain theoretical basis for the application of composite film containing various nanomaterials in food preservation in the future.